# Potentials and pitfalls of permafrost active layer monitoring using the HVSR method: A case study in Svalbard

Andreas Köhler[1] and Christian Weidle[2]

[1]Department of Geosciences, University of Oslo, Post Box 1047, 0316 Oslo, Norway
[2]Institute of Geosciences, Christian-Albrechts-Universität zu Kiel, Kiel, Germany

**Correspondence:** Andreas Köhler (andreas.kohler@geo.uio.no)

**Abstract.** Time-lapse monitoring of the sub-surface using ambient seismic noise is a popular method in environmental seismology. We assess the reliability of the Horizontal-to-Vertical Spectral Ratio (HVSR) method for monitoring seasonal permafrost active layer variability in northwest Svalbard. We observe complex HVSR variability between 1 and 50 Hz in the record of a temporary seismic deployment covering frozen and thawed soil conditions between April and August 2016. While strong variations are due to changing noise conditions, mainly affected by wind speed and degrading coupling of instruments during melt season, a seasonal trend is observed at some stations that has most likely a sub-surface structural cause. A HVSR peak emerges close to the Nyquist frequency (50 Hz) in beginning of June which is then gradually gliding down, reaching frequencies of about 15–25 Hz in the end of August. This observation is consistent with HVSR forward-modeling for a set of structural models that simulate different stages of active layer thawing. Our results reveal a number of potential pitfalls when interpreting HVSRs and suggest a careful analysis of temporal variations since HVSR seasonality is not necessarily related to changes in the sub-surface. In addition, we investigate if effects of changing noise sources on HVSRs can be avoided by utilizing a directional, narrow-band (4.5 Hz) repeating seismic tremor which is observed at the permanent seismic broadband station KBS in the study area. A significant change of the radial component HVSR shape during summer months is observed for all tremors. We show that a thawed active layer with very low seismic velocities would affect Rayleigh wave ellipticities in the tremor frequency band. We compile a list of recommendations for future experiments, including comments on network layouts suitable for array beamforming and waveform correlation methods that can provide essential information on noise source variability.

## 1 Introduction

Environmental seismology is becoming an increasingly popular tool to study earth surface processes and to monitor medium changes in the shallow sub-surface through ambient seismic noise analysis (Larose et al., 2015). The latter approach is often based on noise cross-correlation between two receivers which allows the estimation of the medium's Green's function under the condition of a random seismic noise source distribution in time and space (Shapiro and Campillo, 2004; Sabra et al., 2005). Continuous seismic noise records therefore do not only allow the inversion of sub-surface structures, but also to measure temporal changes therein using seismic noise interferometry (Sens-Schönfelder and Wegler, 2006, 2011; James et al., 2017).

An alternative and well-established single-station approach that makes use of ambient seismic noise is the Horizontal-to-Vertical Spectral Ratio (H/V spectral ratio or HVSR) technique (e.g., Nakamura, 1989; Lunedei and Malischewsky, 2015; Sánchez-Sesma, 2017, and references therein). Peaks in the HVSR curve are related to strong sub-surface seismic velocity contrasts, with shallower interfaces producing higher peak frequencies. The spectral ratio can be inverted for the shallow sub-surface structure based on the diffuse wavefield assumption (García-Jerez et al., 2016; Sánchez-Sesma, 2017) or by interpreting it as representing the frequency-dependent Rayleigh wave ellipticity (e.g., Parolai et al., 2005). HVSRs have been shown to be applicable in a wide range of settings, mostly for measuring site resonance frequencies (e.g., Lachet and Bard, 1994) and mapping sediment thickness, but also more recently in the cryosphere to measure ice properties (Lévêque et al., 2010), glacier and icesheet thickness (Picotti et al., 2017; Yan et al., 2018), or sub-marine permafrost depths (Overduin et al., 2015). Similar to noise interferometry, the HVSR method does in theory allow time-lapse monitoring of the medium below the station, given that the structural change is significant, a source effect can be ruled out, and the Rayleigh wave ellipticity (or diffuse wavefield model parameters) can be extracted precisely enough from the spectral ratios.

It is well-known that a seasonally-frozen shallow surface layer can affect the site response measured through HVSRs (Xu et al., 2010; Cox et al., 2012). Guéguen et al. (2017) for example reported a several day-long HVSR amplitude decrease between 2 and 10 Hz during an air temperature drop below zeros degrees in Grenoble, France. Furthermore, more recently, a few studies interpreted seasonal changes and emerging peaks in HVSRs at higher frequencies as being the result of the thaw-freezing cycle of the permafrost active layer (Abbott et al., 2016; Kula et al., 2018). HVSRs therefore could bear the potential to become a low-cost, passive, and non-invasive method for long-term monitoring of permafrost with high temporal resolution. However, due to the lack of calibration experiments in the field, up to date no standard procedure has been established for such an approach. More studies are needed to explore its limitations and general applicability. For example, a potential pitfall is interpreting HVSR variability as structural change when it is actually due to changes in external site conditions such as noise source distribution and/or meteorological parameters (Chatelain et al., 2008). Violation of the assumption of stationary noise sources might be avoided by using repeating and localized seismic sources, similar to repeating earthquakes that are being used for coda wave interferometry (Snieder, 2006). Environmental seismological research has identified a vast amount of such sources (Larose et al., 2015), e.g., river noise (Burtin et al., 2011), tremors in the cryosphere (Bartholomaus et al., 2015), and anthropogenic structures (Saccorotti et al., 2011; Neuffer and Kremers, 2017).

In this study we explore the potential of the HVSR method for permafrost active layer monitoring using continuous seismic noise records of several months from a temporary seismic deployment close to Ny Ålesund on the Arctic archipelago of Svalbard (Fig. 1). We analyze and compare observed seasonal HVSR variability with forward-modeled changes expected from a thawed soil layer using the diffuse wavefield theory. Furthermore, we analyze HVSR changes of a periodically occurring, localized seismic signal which is present in the record of the permanent seismometer in Ny Ålesund in all available records since 2001. Finally, we discuss the results and compile a list of recommendations for future field experiments from the lessons learned in our study.

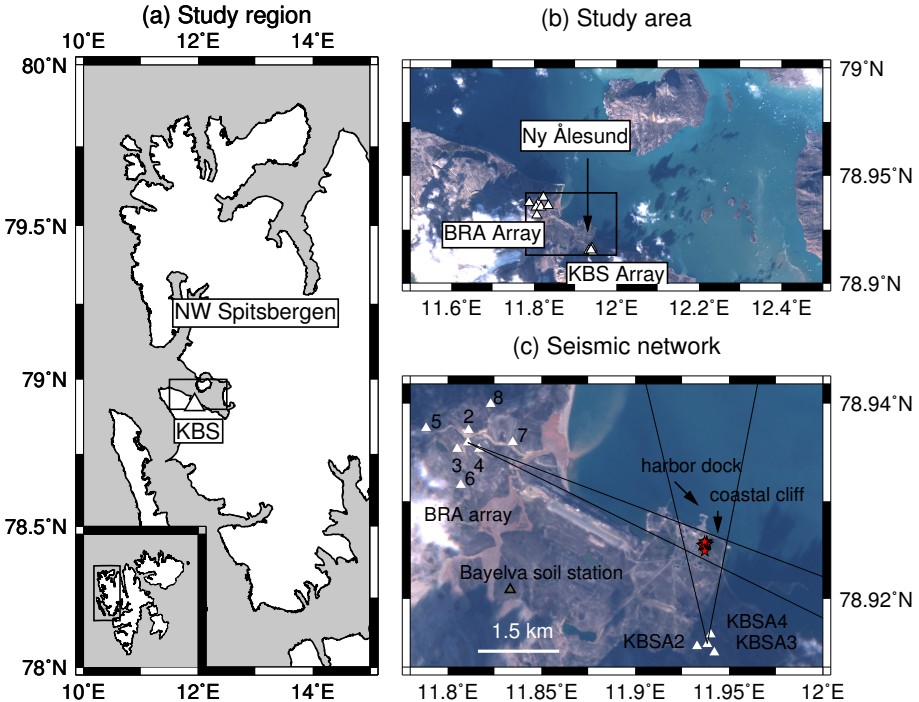

**Figure 1.** Study area, location of instrumentation, and seismic tremor source. (a) Map of northwest Spitsbergen, part of the Arctic archipelago of Svalbard (lower left corner) and location of permanent seismic station KBS. (b) Study area around Ny Ålesund and location of temporary BRA and KBS array. Black rectangle is map section in (c). (c) More detailed location of seismic stations and a coastal cliff with shallow cave shown in Fig. 5d being the source of a repeating seismic tremor (see Section 5). Red stars are tremor locations between April and August 2016. Black lines indicate azimuthal measurement uncertainty when using FK analysis independently on both arrays. Center station of BRA array is BRA1. Numbers indicate the other instrument locations. Background images: Copernicus Sentinel data 2016.

## 2 Data

The permanent Global Seismic Network (GSN) and GEOFON station KBS (network codes IU / GE) is located 1.2 km outside of the settlement of Ny Ålesund (Fig. 1a-b) within a sub-surface, 2 m x 2 m wide, and about 2.5 m deep concrete shelter. Only the channels recording with 40 Hz sampling (BH channels) are used. The 100 Hz data (HH channels) are available in

5    trigger mode only, i.e., solely transient seismic signals unsuitable for noise analysis are being recorded. Between April 12th and September 4th 2016 a temporary seismic network was deployed in the vicinity of Ny Ålesund (Fig. 1b-c). The deployment consisted of two small-aperture seismic arrays built from 11 4.5 Hz three-component geophones connected to Omnirecs DATA-CUBE data loggers, operating with a sampling frequency of 100 Hz. The BRA array (8 stations) was deployed about 2.8 km northwest of the settlement with an inter-station spacing of about 140 m (inner ring) and 500 m (outer ring), and three stations

10    were distributed at about 120 m distance around KBS (KBS array). During installation small holes were drilled into the frozen ground to accommodate the geophone pins. Instruments were covered first with sand and then buried under a rock pile (Fig.

A1). Ground coupling of the instruments degraded during melt season and tilting occurred which increased noise levels in almost all records. The stations were revisited on August 25th. While the three temporary stations of the KBSA array were removed, the coupling and leveling of the BRA array instruments was restored, and data were recorded for 10 more days. Note that the temporary deployment was originally not designed as an active layer monitoring experiment, but for monitoring

iceberg calving at nearby glaciers (Köhler et al., 2016). Similar to most seismic stations (Bonnefoy-Claudet et al., 2006), the seismic noise wavefield measured on our network is mainly composed of ocean micro-seisms at low frequencies ($<1$ Hz) and a mixture of (here limited) cultural noise from the close settlement of Ny Ålesund and effects of local meteorological conditions (wind, ocean swell at local coastline) at high frequencies ($>1$ Hz). Frequent calving activity at nearby tidewater glaciers during summer and autumn (Köhler et al., 2015, 2016) mainly affects intermediate frequencies between 1 and 10 Hz.

**3  HVSRs from ambient seismic noise**

We compute daily-averaged amplitude spectra for the vertical and horizontal components for all stations. Each continuous daily seismic record is divided into 15 minutes long time windows, and the median of the absolute values of the corresponding Fourier spectra is computed. Spectra are smoothed by convolution with a boxcar function (width: 1000 frequency samples with $df$=0.0038 Hz). The horizontal spectra are computed from the North and East component as $\sqrt{\mathrm{North} * \mathrm{North} + \mathrm{East} * \mathrm{East}}$

before computing the spectral ratios. Fig. 2 and 3 show results for a selection of stations together with daily air temperature, soil temperature at 0.39 m depth at a nearby borehole (Boike et al., 2018), and wind speed measured in Ny Ålesund (see Fig. A2 and A3 for rest of stations).

Spectra and HVSRs between April and beginning of September show complex variability. Spectral amplitudes and HVSRs increase strongly in the course of a few days between mid and end of May when air temperatures begin to stay above zero

degrees. This does not happen simultaneously at all stations (e.g., earlier for KBSA2 and BRA2). Furthermore, high wind speed correlates well with high spectral amplitudes during melt season and with short-term HVSR changes (mostly higher amplitude ratios). Stations KBSA2, KBSA4, BRA2, BRA4, and BRA5 show long-term HVSR trends, i.e., a weak, sometimes diffuse, spectral peak apparently gliding from high frequencies (50 Hz) in the beginning of June towards low frequencies in end of August (15–25 Hz). However, wind-related short-term HVSR variability is often stronger than and therefore masking this

long-term trend. At stations KBSA2 and BRA2 the gliding peak trend can be better followed at days of low wind speed. Even if no clear (gliding) peak frequency can be observed over the whole measurement period, stations BRA7 and BRA8 exhibit a strong maximum at 30 Hz for several days during a calm period mid of July (Fig. A2 and A3). Most stations of the BRA array show a clear change in the HVSRs after maintenance on August 25th. For example for BRA2 the gliding frequency peak becomes more pronounced. At BRA1 and BRA4 HVSR amplitudes decrease at all frequencies while at BRA3 (Fig. A2) a new

peak emerges. In addition to the gliding peak at higher frequencies, stations BRA5 and KBSA4 show another weak HVSR peak between 10 and 20 Hz which also seems to have a slight temporal variability in June (decreasing and increasing peak frequency). In contrast to the temporary station, a HVSR peak is observed at KBS close to 20 Hz with amplitudes correlating well with wind speed, however, without clear seasonal variations (Fig. A3).

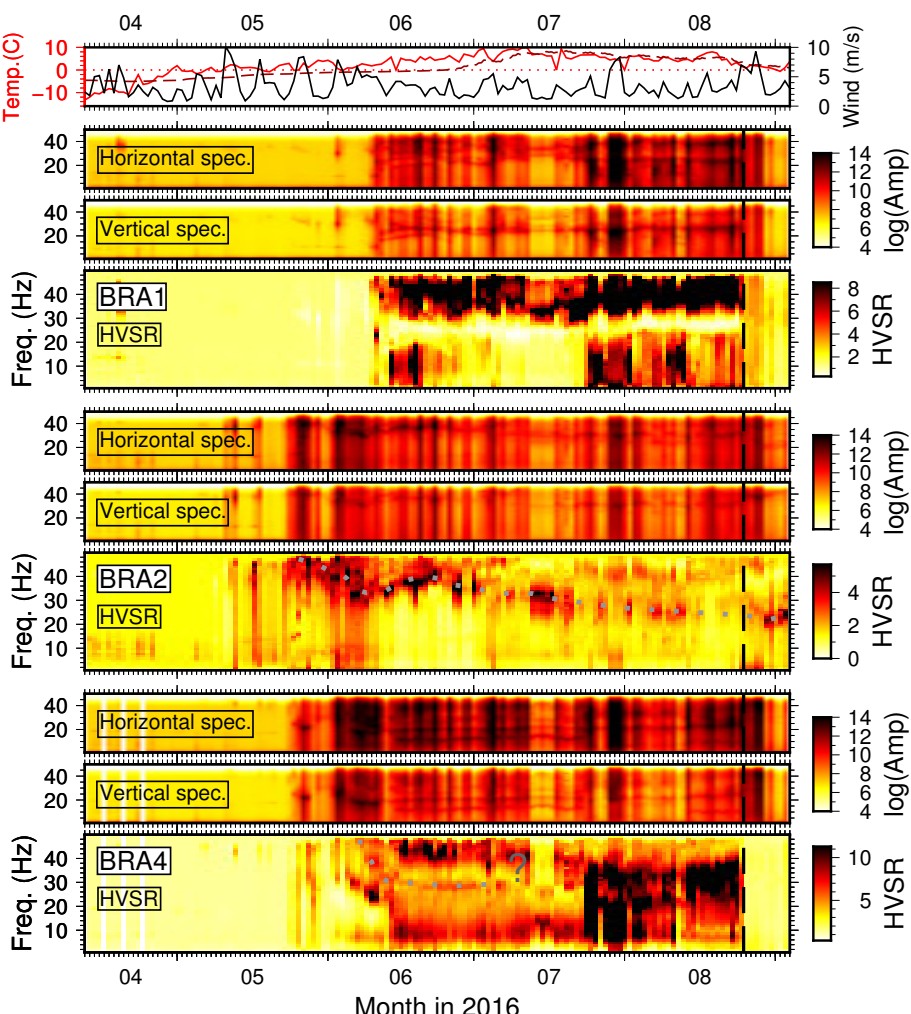

**Figure 2.** Vertical and horizontal component spectral amplitudes and Horizontal-to-Vertical Spectral Ratios (HVSRs) on three stations of the temporary deployment. Dotted lines indicate trend of gliding peak frequencies, question marks ambiguous or unclear peaks, and vertical dashed line date of instrument maintenance (BRA array) or removal (KBS array). Air temperature (red) and daily averaged wind speed (black) measured in Ny Ålesund are shown on top. Dashed dark red line is soil temperature at 0.39 m depth at the Bayelva permafrost observation site (Boike et al., 2018) at 1.6 km distance from BRA and 2.4 km from KBS.

These observations clearly suggest that HVSR variability in our records is complex and cannot merely explained by a single process such as a structural change in the shallow sub-surface. General increase of seismic noise at the onset of and during the melt season is probably mostly due to flowing water and wind. The variability reflects local noise conditions at each individual station affected by topography, vicinity to streams (BRA1, BRA5, and BRA7), exposure to wind, and extent and timing of degrading instrument coupling related to the progress of snow and soil thawing. Stronger correlation with wind speed is probably due to vibration of the instrument loosing coupling which also affects HVSR amplitudes. Hence, HVSRs

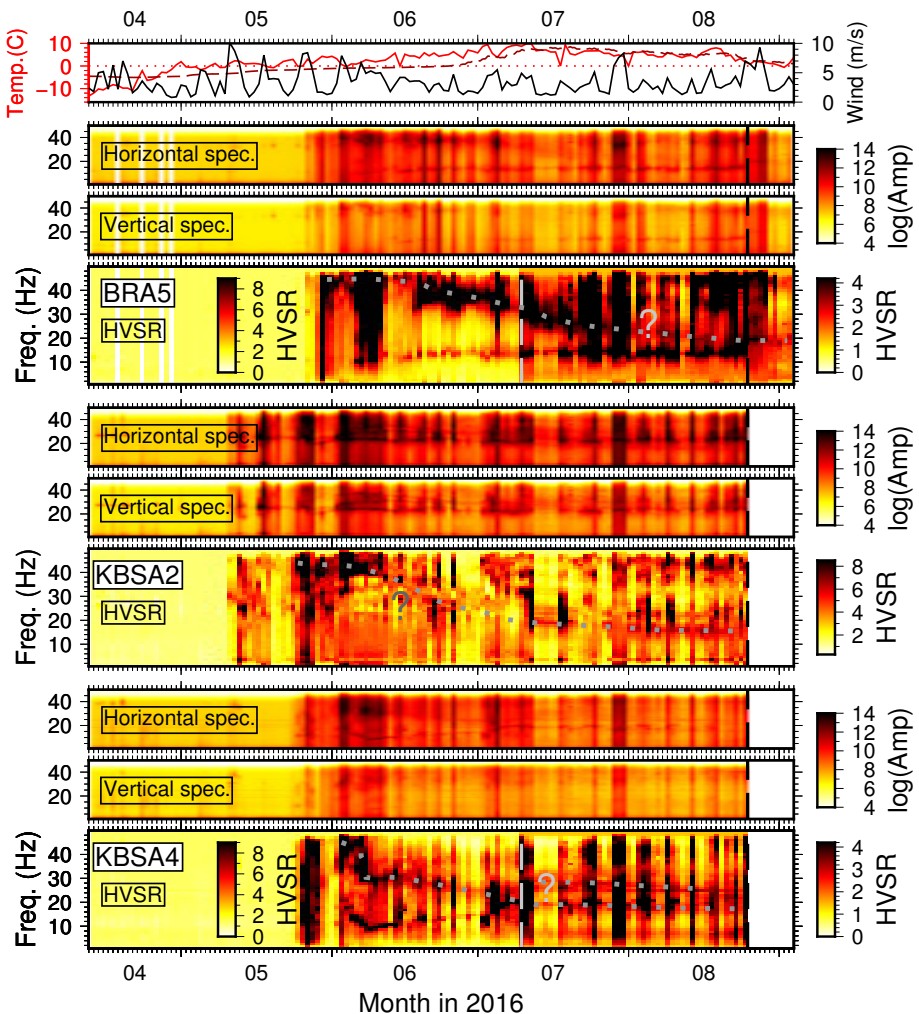

**Figure 3.** Same as Fig. 2 for three more stations. Gray dashed vertical line indicates change of color scale on July 10th. Color scale is clipped at high HVSRs (black) for KBSA4 and BRA5 to enhance visibility of the weak gliding peak at days of low wind speed. The scale used before July 10th is provided to the left.

do not represent the site response during these time periods. The short-term HVSR variability is therefore not related to a structural change and frequency peaks not necessarily to sub-surface interfaces. However, the long-term trend (gliding peak frequency) cannot be easily explained by changing noise conditions and is most likely related to a structural change such as the increasing thaw depth below the station (see discussion below). In fact, the onset of the gliding coincides well with the soil temperature at 0.39 m depth reaching zero degrees. When instrument vibrations dominate and/or ground coupling is too degraded, this structural effect seems to be too weak to be visible during particular time periods or during the entire record for some stations (e.g., BRA1, BRA4). When coupling is restored, strong, non-structural HVSR amplitude peaks disappear (BRA1, BRA4) and/or HVSR peaks presumably due to sub-surface structure are more clearly revealed (BRA2).

**Table 1.** Reference seismic velocity models for the study site based on geological site information available (Haldorsen and Heim, 1999) and adjusted to explain observed Rayleigh wave ellipticities and phase velocities. Winter Model: Frozen active permafrost layer. Summer Model: Unfrozen active layer. HS: Halfspace. Geological units in Haldorsen and Heim (1999): U1: sandstone, U2: shale, U3: chert, glauconitic sandstone, U4: dolomite, limestone, U5: basement. ACL: Thawed active layer.

| Winter model | | | | Summer model | | | | Unit |
|---|---|---|---|---|---|---|---|---|
| Thick. (m) | $V_p$ (km s$^{-1}$) | $V_s$ (km s$^{-1}$) | Den. (g cm$^{-3}$) | Thick. (m) | $V_p$ (km s$^{-1}$) | $V_s$ (km s$^{-1}$) | Den. (g cm$^{-3}$) | |
| | | | | 2 | 1.0 | 0.1 | 1.5 | ACL |
| 90 | 2.5 | 1.0 | 2.0 | 88 | 2.5 | 1.0 | 2.2 | U1/U2 |
| 37 | 3.0 | 1.35 | 2.2 | 37 | 3.0 | 1.35 | 2.2 | U3 |
| 123 | 5.0 | 3.0 | 2.4 | 123 | 5.0 | 3.0 | 2.4 | U3 |
| 350 | 6.0 | 3.5 | 2.7 | 350 | 6.0 | 3.5 | 2.7 | U4 |
| HS | 6.4 | 3.8 | 3.0 | HS | 6.4 | 3.8 | 3.0 | U5 |

## 4 Modeled HVSRs

In order to evaluate the effect of the permafrost active layer, we model HVSRs for a series of sub-surface seismic velocity models using the diffuse wavefield theory, which takes into account surface and body waves (HVInv, García-Jerez et al., 2016; Sánchez-Sesma, 2017). The thaw depth in the Ny Ålesund area can reach up to 2 m in summer (Westermann et al., 2010). The total permafrost depth is between 100 and 150 m (Haldorsen et al., 1996; van der Ploeg et al., 2012). The seismic S-wave velocity change in the active layer is significant ranging from 0.1 to 0.5 km s$^{-1}$ in unfrozen wet soil, depending on liquid water saturation, to 0.9–2.5 km s$^{-1}$ in frozen conditions (e.g., King et al., 1988; LeBlanc et al., 2004; Cox et al., 2012; James et al., 2017). We use a 1D sub-surface velocity reference model (Table 1) inspired by the geological information available (e.g., Fig.4 in Haldorsen and Heim, 1999). We modify the model by introducing an active layer of different thickness (0–2.5 m) and seismic velocity (Vs=0.1–1.0 km s$^{-1}$) to simulate different stages during the thawing process (Fig. 4a-c). The active layer thickness is either fixed and seismic velocity is being decreased step-wise, or the seismic velocity is fixed and the thaw depth is increased successively. The latter model is presumably closer to the real situation, however, there might also be a gradual warming/thawing of the soil from top to bottom leading to a decreasing effective seismic velocity in the active layer over time. In addition, we correct the modeled HVSRs with the instrument response of the geophones to simulate the effect of the anti-aliasing filter at the Nyquist frequency (50 Hz).

As expected, results show the emergence of a HVSR peak related to the increasing or deepening velocity contrast in the shallow sub-surface. The peak frequency decreases to values between 12 and 20 Hz for maximum thaw depths, depending on how low the S-wave velocity is assumed to drop. Spectral ratio amplitudes are affected down to 5 Hz. Due to the upper

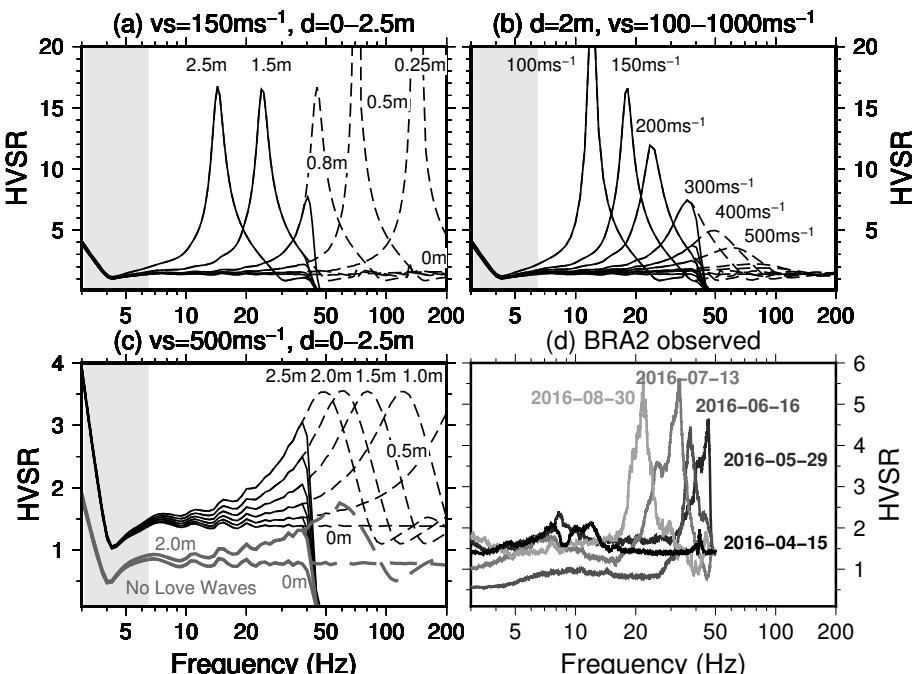

**Figure 4.** (a)–(c) Theoretical Horizontal-To-Vertical Spectral Ratios (HVSRs) modeled using the diffuse wavefield method and sub-surface models of increasing thaw depth $d$ or S-wave velocity $vs$ in the active layer. Reference model in Table 1 is modified accordingly. Black models include Rayleigh, Love and body waves. Gray models in (c) include no Love waves. Gray area indicates tremor frequency band (see Section 5). Dashed curves are modeled HVSRs above Nyquist frequency without using anti-aliasing filter of field instruments. (d) Measured HVSRs at station BRA2 at four different days showing a peak gliding to lower frequencies.

frequency limit at 50 Hz, HVSR peaks begin to emerge below the Nyquist frequency at about 35 Hz, increase in amplitude (Fig. 4c), and then glide towards lower frequencies if S-wave velocity decrease below $0.3\,\mathrm{km\,s^{-1}}$ (Fig. 4b).

The contribution of Love waves in the ambient noise depends on site conditions and affects the amplitude of the HVSR peak, but does in most cases not change the peak frequency itself (Bonnefoy-Claudet et al., 2008). Furthermore, noise source characteristics can lead to variations in the fraction of Love waves (Köhler et al., 2006). In case Love waves are excluded from our forward computation, the HVSR amplitudes are significantly lower compared to the full diffuse wavefield, however, the peak frequency is unaffected (Fig. 4c). The amplitude differences between models including and excluding Love waves is of the same order as amplitude variations for apparent peaks resulting from velocity reduction or thaw depth increase close to the Nyquist frequency.

## 5   HVSRs from a repeating seismic tremor

For better discriminating the causes of HVSR variability, analysis could be restricted to seismic records of a particular localized, repeating, and directional noise source. Furthermore, observations within longer time periods are essential to validate HVSR seasonality observed above. However, since the permanent station KBS has a lower sampling rate, we cannot resolve the relevant frequency range above 20 Hz. Furthermore, since the about 2.5 m deep KBS shelter sits on permanently frozen soil, the effect of active layer variability on HVSRs is expected to become smaller at higher frequencies since decreasing wavelengths sense less of the surrounding medium and more of the concrete shelter. This could explain the lack of a clear HVSR seasonality close to 20 Hz (Fig. A3). However, this might be different if a dominant contribution of seismic signals with longer wavelengths exists. In fact, we observe such a signal at KBS and explore its potential to resolve active layer changes.

### 5.1   The tremor

A characteristic feature at KBS is a pronounced change in the character of ambient seismic noise during certain time periods all year round and in all available records from 2001 until 2016 (except for data gaps between 2001 and 2004). A tremor-like signal occurs, typically lasting for about several hours (Fig. 5a and A4) in a narrow frequency band between 3 and 6 Hz, with a temporally stable spectral peak on the vertical component at 4.5 Hz (Fig. 5c). A remarkably clear semi-diurnal occurrence pattern is observed in the temporal distribution of spectral amplitudes which correlates well with the sea level measured in Ny Ålesund (Fig. 5a). We will refer to this signal as a "repeating tremor" or simply "tremor".

We detect repeating tremors automatically in the entire available KBS record using a short-time over long-time average (STA/LTA) trigger algorithm applied to a time series of vertical component spectral amplitudes (see Appendix B for details). All tremor detections between 2001 and 2016 occur around semi-diurnal tidal maxima in Ny Ålesund. However, during neap tides and low wind speeds, almost no tremors are detected (see average daily wind speed in Fig. 5a). The Fourier transform of the time series of log-spectral powers used for the detector fits remarkably well with the ocean tide spectrum and therefore confirms tidal modulation (Fig. A5). Furthermore, the number of tremors varies seasonally with more detections from late summer until late spring (Fig. 5b).

We use the temporary KBS and BRA arrays to locate tremors which occurred during the deployment period in 2016 by means of frequency-wavenumber analysis (FK, Kvaerna and Ringdal, 1986; Ohrnberger et al., 2004) and the spatial mapping by multi-array beamforming method (SMAB, in supplementary information in Köhler et al., 2016). Figure 1c shows that the tremor source is spatially stationary and very localized at the shoreline in the area of the harbor of Ny Ålesund. Location accuracy is limited because of the resolution limit of array beamforming given the tremor wavelength (about 400 m). A possible source location is a 270 m long and 3–4 m high cliff with a shallow cave-like opening at 200 m distance to the east of the harbor (Fig. 5d). Another potential source is the harbor dock, a grounded artificial structure with an extent of about 100 m. However, ocean wave activity should cause vibration of the dock at high as well as at low tides, unless an unknown mechanism causes vibrations only if the water level reaches the upper part of the structure. We therefore have more evidence for the cliff at the marine cave being the source of the tremor. A reasonable source mechanism for the tremor signal is therefore slamming of

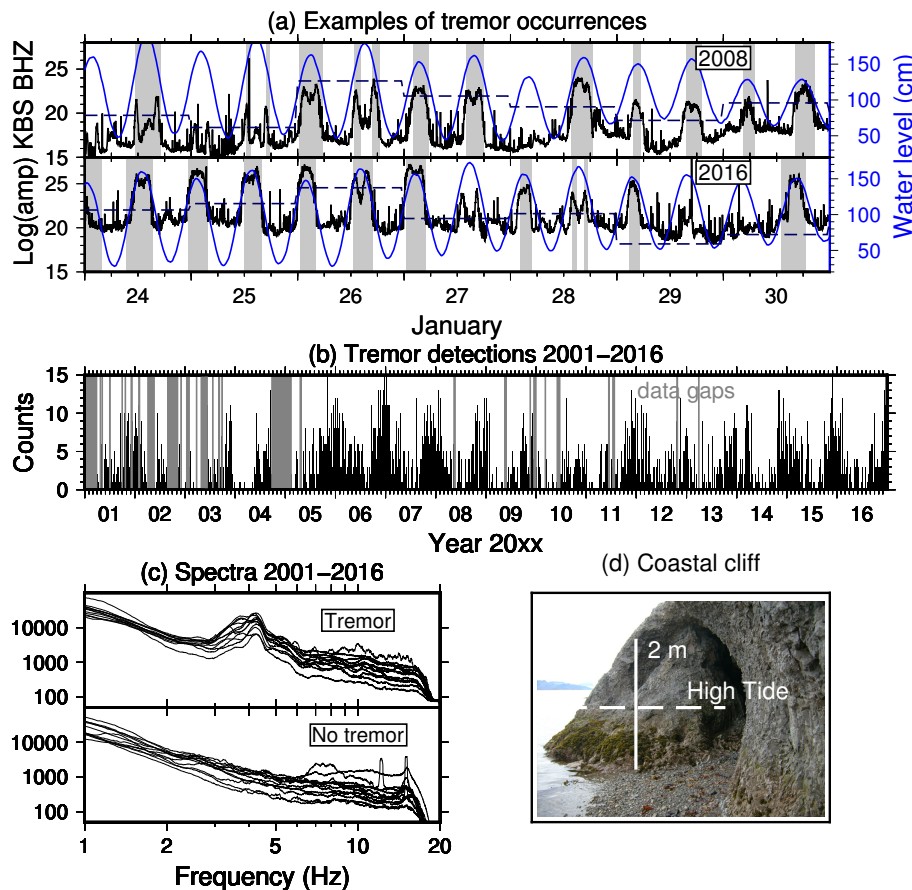

**Figure 5.** Repeating seismic tremor measured at KBS. (a) Temporal distribution of spectral amplitude between 3.4 and 5.7 Hz and water level (chart datum) end of January in 2008 and 2016. High spectral power lasting several hours are tremor time periods which correlate with ocean tides. Gray areas indicate automatic tremor detections. Horizontal dashed lines show relative change in daily wind speed. (b) Temporal distribution of seismic tremor detections. (c) Monthly averaged amplitude spectra of seismic tremor detections (vertical component) and of a selection of monthly time periods without tremors (2016 only). (d) Suggested tremor source: Coastal cliff with shallow marine cave (Fig. 1c).

breaking sea waves at the cliff during high tides and significant ocean wave activity (Adams et al., 2002; Young et al., 2016), often accompanied by high wind speeds. At low tides and/or at high tides during the neap tide cycle, a narrow beach is exposed and the ocean waves do not reach the cliff which explains the temporal distribution of tremor occurrences. Furthermore, ocean wave activity usually being stronger during autumn and winter and spring tides being strongest around the equinox in March and September, is a good explanation for the seasonality (Fig. 5b). Our observations are consistent with previous studies on ocean wave cliff interaction causing microseismic cliff-top ground motion within a frequency band of 1 to 50 Hz (Dickson

and Pentney, 2012; Norman et al., 2013) with peaks around 10 Hz (Jones et al., 2015; Earlie et al., 2015) and tidal modulation (Earlie et al., 2015). The slamming forces of breaking ocean waves might be stronger in the cave because of the confined space, which could be an explanation for the signal strength even at 2 km distance (BRA array). No similar signals are observed from a few other shoreline cliffs in the area which are located between mostly flat beaches.

Beamforming analysis of the vertical components of the KBS array suggests that the tremor signal consists predominantly of surface waves. Apparent seismic phase velocities show typical dispersion with values between 1.5 and 2.0 km s$^{-1}$ (Fig. A6b). In contrast to frequencies below 2 Hz and above 6 Hz where ambient seismic noise dominates the wavefield, the back-azimuth in the tremor frequency range fluctuates only slightly and points clearly to north on average (Fig. A6a).

## 5.2   Tremor spectrum and polarization

We analyze all three spatial wavefield components to gain more insights into the propagation properties of the seismic tremor. Figure 6b shows the spectral amplitudes of the radial component for a single tremor testing different back-azimuth angles. The spectrum is computed as the median of individual amplitude spectra obtained for 15 minutes long time windows. The first and last 35 minutes, where the tremor gradually emerges or disappears, are not analyzed to prevent ambient seismic noise affecting the results. The following results are representative for all other tremors between 2001 and 2016. It is striking that high spectral

amplitudes on the horizontal components alternate between the frequency ranges 3–4 Hz and 4–5 Hz for different back-azimuth angles, whereas on the vertical component the entire frequency range 3–5 Hz dominates (Fig. 6a). Maximum amplitudes in both frequency bands correspond to perpendicular directions which do not coincide with the propagation direction from north to south as inferred from vertical component FK analysis. In fact, the maximum between 4 and 5 Hz is about 40 degrees off the propagation direction.

We evaluate the tremor polarization by cross-correlating the vertical and the Hilbert-transformed radial component. In case of dominant surface waves, the radial component for a back-azimuth of zero degrees (location of tremor source) should yield a pure Rayleigh wave with elliptic polarization. However, according to Fig. 6c, the polarization maximum is clearly shifted towards positive back-azimuth angles between 4 and 5 Hz coinciding well with the radial component amplitude maximum. On the other hand, correlation of vertical and Hilbert-transformed radial component between 3 and 4 Hz and thus ellipticity is very

low for all backazimuth angles. This suggests that Rayleigh waves on the horizontal components only dominate between 4 and 5 Hz for an (apparent) back-azimuth of about 40 degrees. Furthermore, it seems that Love waves from the same direction dominate between 3 and 4 Hz since maximum amplitudes are observed for a rotation angle of 130 degrees, the corresponding transverse component. The lack of Rayleigh wave energy on the radial component in this frequency band and the presence on the vertical component can be explained by a trough in the frequency-dependent ellipticity. It remains, however, unclear why

Love waves disappear between 4 and 5 Hz.

    The back-azimuth discrepancy between vertical FK and polarization analysis may be due to azimuthal anisotropy or a mis-orientation of the KBS instrument. The latter possibility can be excluded since systematic bias towards positive back-azimuth angles is also observed on the temporary stations of the KBS array. Furthermore, an analysis of P wave polarization from regional earthquakes at KBS revealed a similar behavior. There is a systematic, back-azimuth dependent bias at KBS

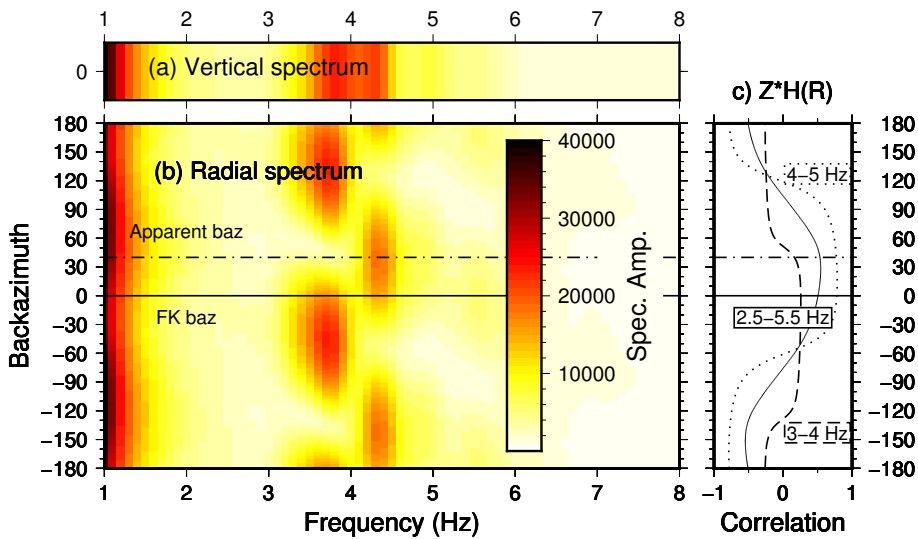

**Figure 6.** (a) and (b) Example amplitude frequency spectra at KBS for tremor occurring between 2016-05-12T03:02:00 and 2016-05-12T06:17:00 for vertical and radial component assuming different back-azimuth values. (c) Correlation coefficient between vertical and Hilbert-transformed radial component assuming different back-azimuth values in different frequency bands. High values are expected in case of Rayleigh waves on the radial component. Tremor backazimuth from vertical FK analysis (FK baz) and apparent backazimuth corresponding to maximum correlation between 4 and 5 Hz (Apparent baz) are indicated. Discrepancy is probably due to azimuthal anisotropy.

between polarization angle and expected back-azimuth (Fig. A7). This bias is positive at zero degrees back-azimuth. Sub-surface geology in the Ny Ålesund area exhibits southwest dipping sediment layers (Fig.3 and 4 in Haldorsen and Heim, 1999) which could give rise to azimuthal anisotropy, i.e., a rotation of the polarization ellipsoid (clockwise from north) with respect to propagation direction (north to south). A quantification and further analysis of this finding is beyond the scope of this paper

and should be subject of future studies.

### 5.3 Variability of Rayleigh wave ellipticity

We compute HVSRs of all tremor records at KBS to analyze the Rayleigh wave ellipticity using the same processing as for the ambient noise. Since we found clear evidence that the angle separating Rayleigh and Love waves on the radial and tangential components does not coincide with the propagation direction inferred from the vertical component (Fig. 6) and as suggested by

the tremor source location, we compute the radial to vertical (RVSR) spectral ratios using a back-azimuth of 40 degrees. Figure 7b shows that the RVSRs are very stable and their standard deviations low within the tremor frequency band. A complex peak-trough shape of the RVSR curve is revealed. After testing different (1D) sub-surface velocity models based on our reference model (Table 1), it turned out that this behaviour can only be explained by a mixture of fundamental and higher mode Rayleigh wave ellipticities (compare Fig. 7a and b). The first trough at 4 Hz can be related to the ellipticity minimum of the fundamental

and first higher mode. The fundamental mode peak below 3 Hz lays outside the tremor band and is probably therefore not

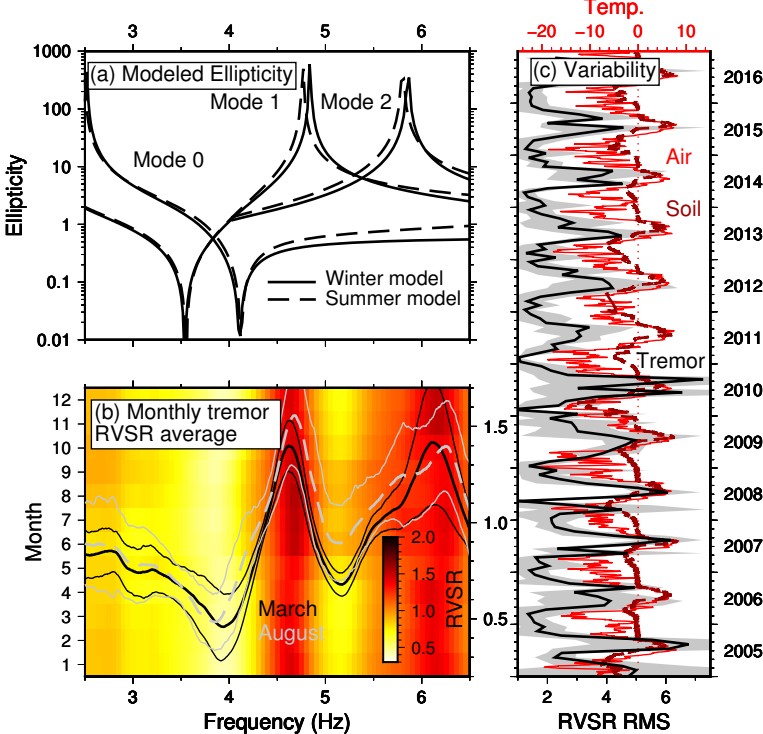

**Figure 7.** (a) Rayleigh wave ellipticities for fundamental and two higher modes in the tremor frequency band computed from the reference model and modified model by introducing a 2 m thick top low-velocity layer supposed to represent a thawed active permafrost layer. (b) Temporal variation of tremor RVSRs averaged over individual months and years 2010-2016. Average RVSRs for March and August and standard deviations show that the seasonal change in amplitude is significant and consistent ($p < 0.01$ between 4.0 and 5.8 Hz for Welch's T-test). (c) Air temperature measurements in Ny Ålesund (10 day running average), soil temperature at 0.59 m depth (dark red dashed) (Boike et al., 2018), and monthly averaged RMS difference for frequency range 4.0–5.5 Hz between averaged RVSRs in February 2016 and each tremor RVSR. Standard deviations are shown as gray areas. Years 2001–2004 are not shown because of long data gaps.

revealed. The first RVSR peak between 4 and 5 Hz seems to coincide with the first higher mode ellipticity maximum. The next trough would then be related to an ellipticity minimum which results from the superposition of first and second higher mode. At the upper limit of the tremor band at 6 Hz another peak could be related to the second higher mode peak.

The radial component HVSRs of all tremor occurrences between 2001 and 2016 exhibit very similar shapes (monthly averaged RVSRs are provided in the supplement S03). However, there is a slight, but significant ($p < 0.01$ for equal mean hypothesis in Welch's T-test) seasonal variation in the amplitudes between 4.0 and 5.8 Hz (Fig. 7b). The amplitudes are higher during the summer months between June and September. We quantify the RVSR variability by computing the RMS difference between 4.5 and 5.5 Hz with respect to the average RVSR of tremor records in February 2016 (Fig. 7c) which reveals a clear seasonality in all years. As soon as air and ground temperatures increase above zero degrees, RMS values increase rapidly, before dropping again in autumn when temperatures approach negative degrees.

## 6 Discussion of the reliability of HVSRs for permafrost monitoring

The results of our field measurements and theoretical modeling reveal a number of challenges and pitfalls when attempting to use HVSRs to monitor the active permafrost layer. In case of ambient seismic noise, the general broadband HVSR amplitude increase and the emergence of amplitude peaks in the beginning and during the melt season could be mistaken for a direct structural effect of the active layer. Furthermore, strong HVSR peaks resulting from short-term changes in the noise sources, e.g. wind affecting the instrument directly or generating noise at or at close proximity to the measurement site, could be misinterpreted as HVSR peaks related to sub-surface interfaces if the recording period is too short or wind speed measurements are not available. At the same time, a weaker structural peak might be masked by such noise sources. Moreover, an emerging HVSR peak close to the Nyquist frequency could be an artifact of the instrument anti-aliasing filter (Fig. 4c), i.e., it could be related to an emerging peak at higher frequencies and would lead to an overestimation of the thaw depth if the apparent peak is misinterpreted. Furthermore, a frequency-dependent seasonal change of the relative contribution of Rayleigh and Love waves will affect HVSR amplitudes and could give rise to misinterpretation of the caused HVSR variability that is not related to a structural change. Finally, for measuring HVSR changes caused by the active layer, seismic instruments have to be deployed on top of or inside the soil which naturally leads to degrading coupling, tilt, and/or instrument vibrations during thawing. The processes above include issues known from previous studies to affect HVSRs. For example, Chatelain et al. (2008) mentioned among other effects strong tilt, strong wind when recording next to a feature connected to the ground, and heavy rain. The main focus of Chatelain et al. (2008) was the frequency range below 20 Hz, however, one would expect these issues to become even more relevant at higher frequencies, a reason why it was recommended to restrict HVSRs analysis to frequencies below 10 Hz. Nevertheless, in order to resolve a HVSR peak caused by the active layer, we need to take these frequencies into account.

Another finding of Chatelain et al. (2008) are strong effects related to the nature of the shallow uppermost layer. Thick (>10-15 cm) mud, ploughed and/or water-saturated soil, was shown to lead to higher HVSR amplitudes and appearance of artificial peaks at higher frequencies. Similar, we have clear indications for a shallow, structural variation causing a temporal change in the HVSRs at 5 out of 11 seismic stations and short-term HVSR peaks at two more stations during days of low wind speed that can be attributed to the permafrost active layer (Fig. 8). The gliding frequency peaks are consistent with a realistic active layer thawing process starting in beginning of June and reaching consistently with the modeling results a thaw depth of about 2 m and S-wave velocities between 0.15-0.25 km s$^{-1}$ at the end of the summer. The best example is station BRA2 where a peak emerges in May at 46 Hz (probably underestimated because of the anti-aliasing filter) from a flat HVSR curve measured in April (Fig. 4d). Subsequently, the peak frequency decreases to 38 Hz in June, 33 Hz in July, and 22 Hz in August. Furthermore, HVSR peak amplitude ratios lay in the range of the modeled values. BRA2 was located at the eastern foot of a small hill, probably shielding the instrument more efficiently from wind coming dominantly from West. Hence, our results suggest that HVSRs can indeed be used to monitor the thawing-freezing cycle in permafrost, given that a careful analysis of the temporal variability has been carried out as pointed out above. However, more calibration experiments are necessary to relate peak frequency directly to thaw depth and soil properties, as well as to identify preferable sites for such measurements.

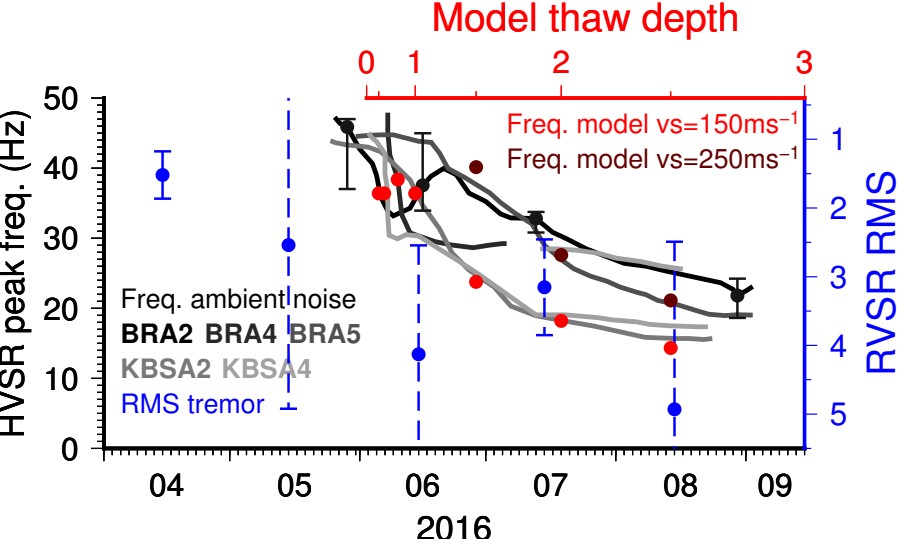

**Figure 8.** Effect of permafrost active layer on HVSR measurements. Comparison of observed ambient noise HVSR peak frequencies for stations BRA2, BRA4, BRA5, KBSA2, and KBSA4 (solid lines) and modeled peak frequencies (red and dark red symbols) taking into account anti-aliasing filter of seismic instrument. For station BRA2 additional black symbols and error bars show peak frequencies and uncertainties corresponding to days in Fig.4d. For the x-axis on top representing modeled thaw depth, we assume a square root dependency with time from beginning of June. Tremor RVSR RMS values from Fig.7 are shown. RMS and peak frequencies follow similar trend.

As a special case of the known seasonal effect on HVSRs related to the thawing-freezing cycle (e.g., Guéguen et al., 2017), variability caused by the permafrost active layer has been reported previously (Abbott et al., 2016; Kula et al., 2018). Instead of geophones, Abbott et al. (2016) (same experiment as James et al., 2017) used Posthole sensors buried in the active layer since these instruments are less sensitive to tilt. Such an instrumentation would therefore eliminate some of the noise issues
we face with our deployment. Furthermore, in that study emerging HVSR peaks between 10 and 30 Hz were observed during summer, which, however, could not be explained by the relatively shallow active layer thickness of 68 cm at their study site. Kula et al. (2018) described seasonal HVSR variability at a seismic station in southern Svalbard. Since a permanent station was used with 100 Hz sampling, higher frequencies were being resolved than possible at KBS, and instrument coupling was not an issue. However, similar to our results, the authors acknowledged that low-frequency HVSR peaks (e.g., at 12 Hz) and
overall seasonal HVSR amplitude increase is due to wind noise and/or human activity at the research station in summer. They also described a peak, but not gliding as in our case, emerging in June at 40 Hz close to the Nyquist frequency accompanied by a minimum at 30-35 Hz which they attribute to active layer thawing. The observations of both previous studies support our conclusion that HVSR interpretation must be done carefully as strong HVSR peaks or amplitude increases in general are not necessarily related to shallow structural changes, although they appear seasonally.
A station network allows us to pursue different approaches than simply applying the single-station HVSR method. Beside two-station noise interferometry to measure seismic velocity changes (James et al., 2017), array analysis makes it also possible

to measure the frequency-dependent ratio of Rayleigh and Love waves on the horizontal components (3c-MSPAC, Köhler et al., 2007) and to analyze noise directionality through array beamforming (Ohrnberger et al., 2004). However, the minimum inter-station spacing must be carefully adapted to the frequency range to be resolved. Since our array geometries were designed to detect and locate calving events between 1 and 10 Hz, we cannot use these array methods due to spatial aliasing and lacking wavefield correlation at frequencies higher than 10 Hz. A more adequate station setup would potentially allow differentiating between effects of changes in Love wave contribution, noise sources, and propagation medium on HVSR variability. We tried ambient noise interferometry between our array stations as well. However, we encountered lack of waveform correlation due to too large inter-station distances and locally uncorrelated noise at frequencies higher than 10 Hz. Hence, no seasonal velocity changes related to the active layer could be measured such as successfully done by James et al. (2017).

Utilizing a localized and repeating seismic signal for permafrost monitoring might be an alternative to ambient noise HVSRs. The seasonal variations observed in our tremor RVSRs could be either due to changes in the propagation medium or the tremor source itself. In general, the HVSR method is supposed to remove source effects. In our case for example, the tremor source magnitude variability should affect the vertical and radial component of the Rayleigh wave measured at KBS in the same way. However, we cannot fully exclude the possibility that noise not related to the tremor increases stronger on the horizontal components during summer than on the vertical component. If the RVSR variability is due to medium changes, the active permafrost layer is a good candidate to explain our observations, though the strongest amplitude increase is expected at much higher frequencies (Fig. 4). Nevertheless, modeling Rayleigh wave ellipticities shows that the tremor frequency band is slightly affected. We obtain a clear increase in ellipticity for the first and second higher mode above 4.5 Hz for a model assuming very low S-wave velocities in the active layer (Table 1, Fig. 7a). This is consistent with Guéguen et al. (2017), who observed a significant HVSR amplitude change within the same frequency band (2–10 Hz) caused by a 0.75 m deep frozen layer. However, we cannot exactly reproduce our measured RVSR change due to lacking knowledge about the relative contribution of Rayleigh waves modes and possibly body waves, as well as probably deviations from a 1D sub-surface structure that exist due to dipping layer in the study area (Haldorsen and Heim, 1999). Modeling ellipticities using 2D or 3D structure might help to better explain our observations. The presence of a repeating, localized tremor signal at higher frequencies around the HVSR peak directly related to the unfrozen layer in summer, would allow us to asses the seasonality with higher certainty through directly measuring the peaks frequency change. This potential has to be followed up by more related studies in future.

In our case, ambient noise and tremor HVSRs complement each other. The gliding HVSR peak frequency can only be measured from a short record (temporary network), while a long-term record is available for KBS to analyze inter-annual variability. However, since a permanent station within a shelter structure such as KBS might not be sensitive enough to active layer variability at high frequencies or has a too low sampling frequency, signals with longer wavelengths are needed. Analyzing the tremor signal allows measuring HVSR variability at lower frequencies that would otherwise (i.e., with ambient noise) not be sensitive enough to resolve active layer thawing. Although the measured quantities are different, ambient noise HVSR peak frequencies and tremor HVSR RMS values exhibit consistent variability during the measurement period (see Fig. 8), presumably related to the same cause, i.e., the active permafrost layer.

## 7 Summary and recommendations

We apply the HVSR method to a temporary seismic deployment and the permanent station KBS in northwest Svalbard to investigate its applicability for permafrost active layer monitoring. As expected, ambient noise HVSR variability is strongly affected by changing external site conditions but also reveals a seasonal trend. A gliding peak frequency between 50 and 15 Hz is observed that most likely indicates a deepening thaw depth from June until September as confirmed by modeled HVSRs using the diffuse wavefield assumption. Furthermore, we describe a repeating, ocean swell and tide related seismic tremor in the record of KBS. We are able to extract the frequency-dependent ellipticity from the tremor radial-to-vertical spectral ratios. We find a significant seasonal variation between 4.5 and 5.5 Hz. Although these frequencies are less sensitive to shallow medium changes, we show that Rayleigh wave ellipticities are still affected by the thawed permafrost active layer.

Our results demonstrate that active layer monitoring would benefit from more purpose-built seismic networks and that interpretation of spectral ratio variability must be done carefully to exclude non-structural effects. We confirm previous, general recommendations and known issues of the HVSR method (Chatelain et al., 2008), which become even more important at the high frequencies needed to resolve the active layer HVSR peak. In summary, we suggest the following recommendations, including and emphasizing those given previously and being of special relevance for future passive seismic experiment that have the goal to measure permafrost active layer variability:

1. The seismic sampling rate should be at least 200 Hz to capture HVSR peaks of shallow, emerging interfaces and to avoid misinterpretation of apparent peaks close to the Nyquist frequency.

2. If logistically feasible, repeated maintenance at temporarily deployed instruments during the melt season is strongly recommended to keep ground coupling stable. Digging instruments deeper into the soil (if deployment is done during thawed conditions) and/or using Posthole sensors if affordable is an alternative (Abbott et al., 2016). Cementing the sensor a few decimeters below the surface on a small plate might be another option (Chatelain et al., 2008).

3. A careful evaluation of HVSR variability caused by non-structural effects (e.g., Chatelain et al., 2008) must be performed, for example using co-located wind speed measurements. As noted in previous studies, time periods with strong wind noise should be excluded from analysis and/or an efficient wind shielding should be used.

4. The deployment of small-aperture seismic arrays with minimum 4 elements and with minimum inter-station distances not larger than 5 to 10 m is recommended to allow:

    (a) Measuring the frequency- and time-dependent contribution of Rayleigh and Love waves at high frequencies (3c-SPAC method) since a change would affect HVSR amplitudes (Bonnefoy-Claudet et al., 2008).

    (b) Measuring changing noise source directionality and resulting effects on HVSRs (backazimuth measurements with beamforming / FK analysis).

    (c) Combining HVSRs measurements with seismic noise interferometry (James et al., 2017).

    (d) Comparison and evaluation of HVSRs of close stations affected by more similar local noise and ground conditions.

5. Making use of repeating directional noise sources if applicable has the potential to avoid source variability affecting the HVSRs. If the frequency content of such a source is too low, temporal HVSR increase might still be connected to a peak at higher frequencies. In addition, a purpose-built linear seismic array aligned with propagation direction would allow the application of noise interferometry.

5   HVSR analysis cannot yet be considered to be a stand-alone tool to measure permafrost active layer variability without including seismic expert knowledge and taking into account site-dependent factors. However, our study clearly shows the potential of the HVSR method. We are confident that more case studies, long-term experiments, and improved instrumental set-ups will help to establish this approach as a useful supplementary tool in permafrost research.

*Data availability.* Data of station KBS are freely available through IRIS (Albuquerque Seismological Laboratory (ASL)/USGS, 1988).
10  The seismic record of the temporary network stations will become publicly accessible through the Geophysical Instrument Pool Potsdam GIPP (http://gipp.gfz-potsdam.de/webapp/projects/view/536). Measured sea level data from Ny Ålesund were obtained from kartverket.no. Meteorological data are available from re3data.org (2018) and soil temperatures at station Bayelva from Boike et al. (2017). Copernicus Sentinel data from 2016 was used in Fig.1.

## Appendix A: Supporting figures for HVSRs from ambient seismic noise

Fig. A1 shows examples of deployed seismic sensors. Fig. A2 and A3 show the HVSRs for the stations not shown in the main text.

## Appendix B: Automatic detection and temporal distribution of the repeating tremor

Repeating tremors in the KBS record are detected using a STA/LTA trigger applied to a time series of vertical component spectral amplitudes. We compute the logarithm of spectral power between 3.4 and 5.7 Hz in non-overlapping 150 s long time windows. A STA length of 25 minutes, a LTA length of 25 hours, and a STA/LTA threshold of 1.15 is used. If the threshold is exceeded for a sample (time window), the occurrence of a tremor is declared. Samples are assigned to the same tremor if gaps between exceeded thresholds are shorter than 1 hour. If the gap is longer, the onset of a new tremor is declared. Detections

with duration less than 25 minutes are sorted out. All detection parameters are found by evaluating if clear, visually identified tremors are correctly detected, while minimizing the number of false detections. Visual post-processing is done to reject a few false positives so that only real tremors are used for further processing. The list of all detected tremors is provided in the supplement S02. Tremors were detected around semi-diurnal tidal maxima in Ny Ålesund (Fig. 5), except during neap tides and at low wind speed. Sometimes two tremors are declared if the amplitudes exhibit a two-sided distribution, i.e., peaks at

the start and the end of a tremor (see for example 2008-01-26 and 2016-01-28 in Fig. 5a). The amplitude spectrum of the time series of log-spectral powers used for the detector shows prominent semi-diurnal tidal peaks (Fig. A5, Darwin symbols of tides: $M_2$, $S_2$, $N_2$). Furthermore, diurnal ($K_1$, $O_1$), terci-diurnal ($M_3$), and quarti-diurnal ($M_4$) peaks are clearly revealed. The neap-spring tide cycle (14.75 days, $M_{sf}$) appears as a weak peak in the spectrum. In some years (2003, 2004, 2009–2011) the number of tremor detections drops in the beginning of the year which could be an effect of sea ice preventing ocean wave

activity. Note that in recent years (from about 2013), no land fastened sea ice has been observed at the coast of Ny Ålesund (pers.com. C. Nuth, 2018).

## Appendix C: Supporting figures for tremor spectrum and polarization

Fig. A6 shows results of FK analysis for a tremor record. Measured back-azimuth at KBS array and P wave polarization angle for regional earthquakes are shown in Fig. A7.

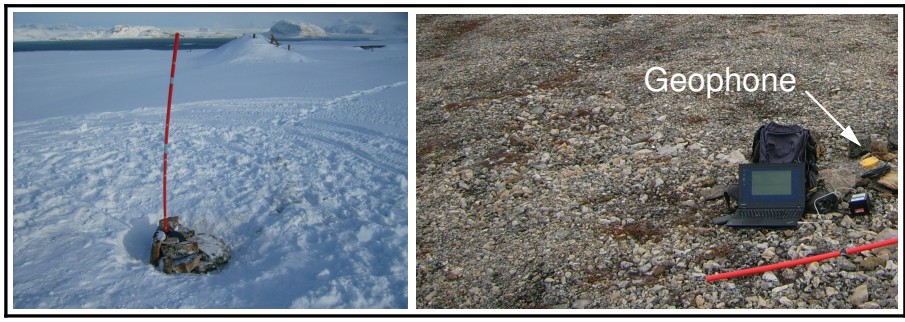

**Figure A1.** Photos of a station of the KBS array after deployment in April and a station of the BRA array during data retrieval in August (geophone uncovered).

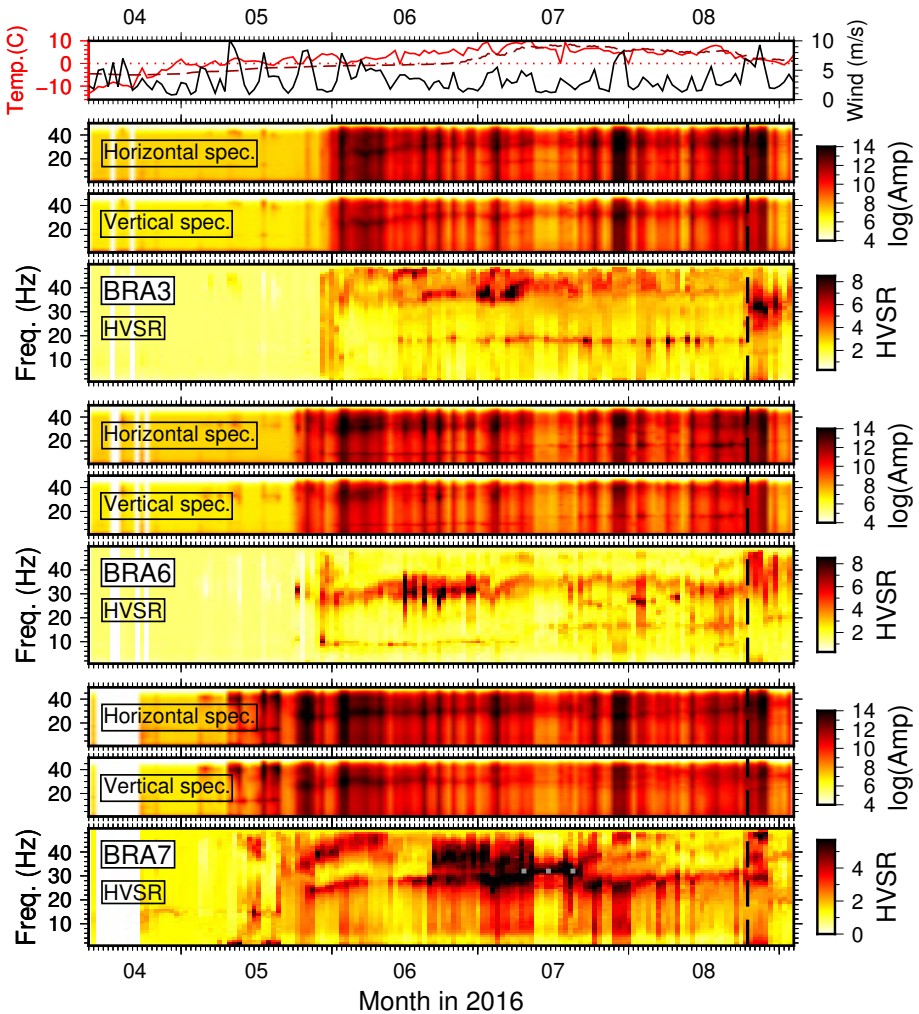

**Figure A2.** Same as Fig. 2 for three more stations.

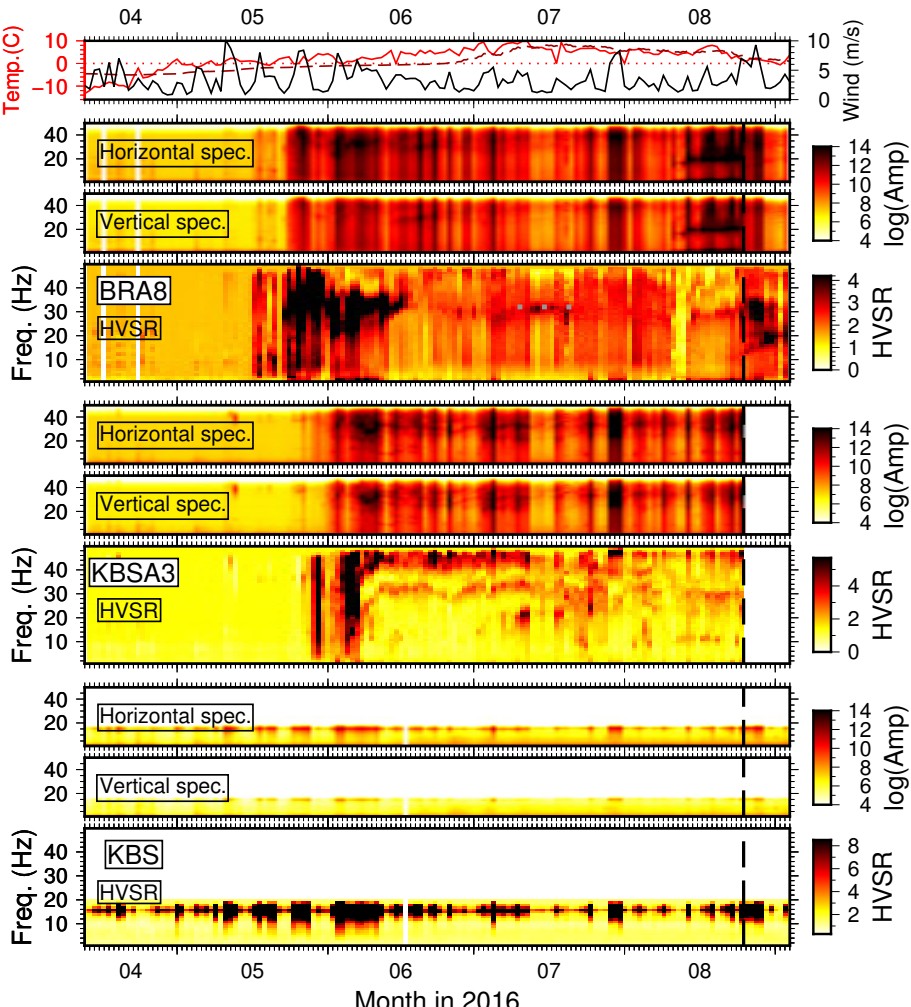

**Figure A3.** Same as Fig. 2 for three more stations.

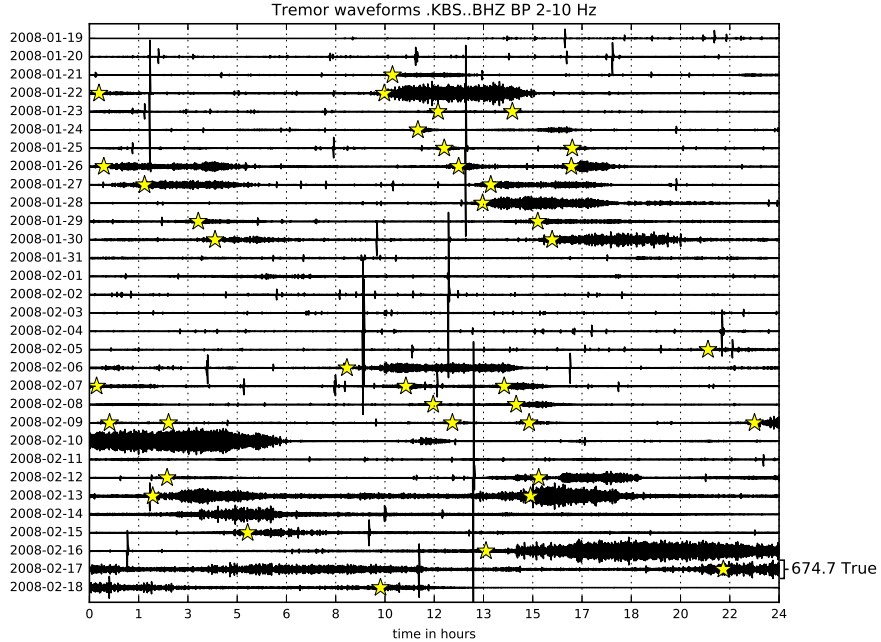

**Figure A4.** Example of repeating seismic tremor waveforms recorded at KBS. Detected tremor onsets are indicated by yellow stars. Waveform data of the tremor on 2008-01-22 are provided in the supplement S01.

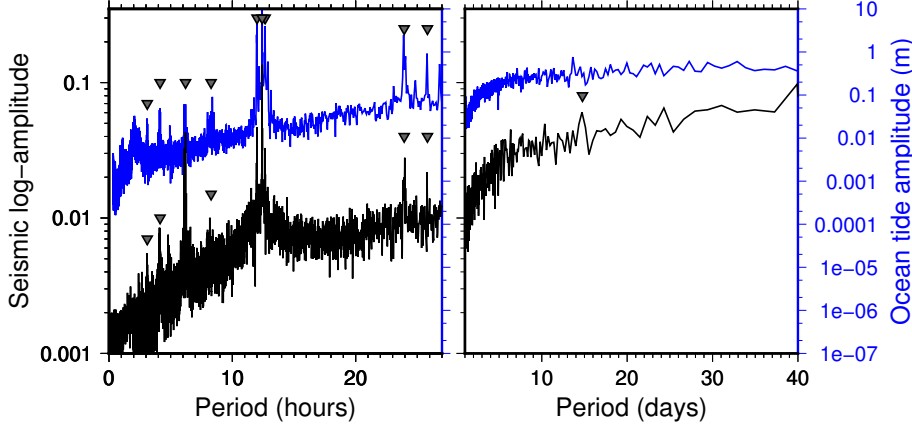

**Figure A5.** Amplitude spectrum of time series of log-spectral powers of KBS vertical component seismic data (see Fig. 5a) and of measured sea level in Ny Ålesund between 2005 and 2016. Gray triangles indicate theoretical ocean tide periods at (from left to right, Darwin symbol of tide in brackets) $3.105\,\mathrm{h}\ (M_8)$, $4.14\,\mathrm{h}\ (M_6)$, $6.21\,\mathrm{h}\ (M_4)$, $8.28\,\mathrm{h}\ (M_3)$, $12.0\,\mathrm{h}\ (S_2)$, $12.42\,\mathrm{h}\ (M_2)$, $12.658\,\mathrm{h}\ (N_2)$, $23.93\,\mathrm{h}\ (K_1)$, $25.82\,\mathrm{h}\ (O_1)$, and $14.75\,\mathrm{days}\ (M_{sf})$.

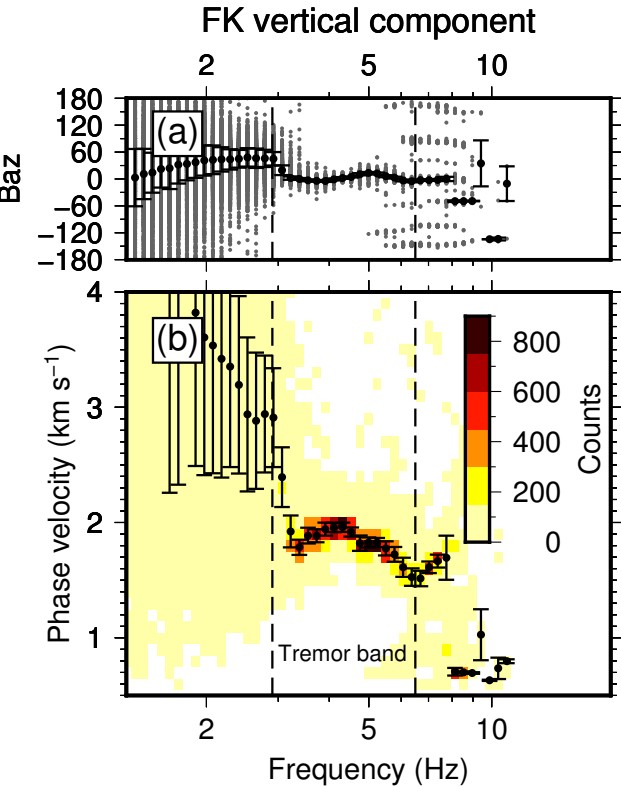

**Figure A6.** Example of frequency-wavenumber (FK) analysis of vertical components of KBS array for tremor occurring between 2016-05-12T03:02:00 and 2016-05-12T06:17:00. (a) All back-azimuth (Baz) measurements at maximum beampower and with coherency (normalized beampower) > 0.7 for 600 s long time windows during tremor occurrence (gray symbols) and median with median deviation (black). (b) Color-coded histogram (counts) of phase velocity measurements for same time windows as in (a) and median with median deviation.

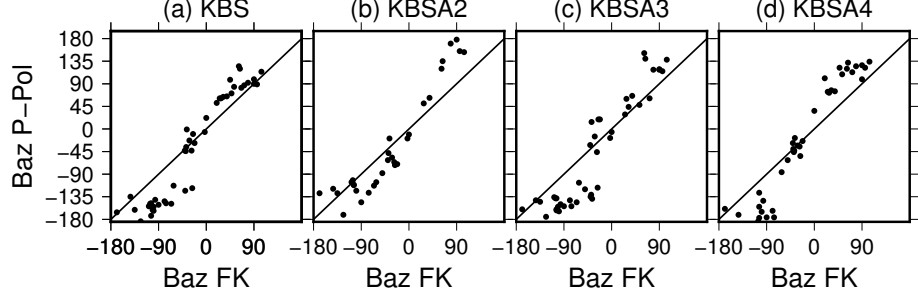

**Figure A7.** Back-azimuth measured with FK analysis at KBS array vs. stations P wave polarization angle measured from regional earthquakes.

*Author contributions.* AK and CW initiated the study. AK processed and analyzed the seismic data and prepared the manuscript. CW was responsible for field instrumentation and assisted in the field experiment and manuscript editing.

*Competing interests.* The authors declare that they have no conflict of interest.

*Acknowledgements.* This study was financed by the Norwegian Research Council funded CalvingSEIS (244196/E10) and SEISMOGLAC
5 (213359/F20) projects. Seismic instrumentation for temporary network was provided by the Geophysical Instrument Pool of GFZ Potsdam, Germany. Special thanks go to Christopher Nuth (PI of CalvingSEIS) for organizing logistics in Ny Ålesund and for helping together with Cesar Deschamps-Berger during instrument deployment. We used ObsPy (Beyreuther et al., 2010) for seismic data analysis. Figures were produced using GMT (Wessel and Smith, 1998). Rayleigh wave ellipticities were computed using Geopsy (http://www.Geopsy.org). We thank Antonio Garcia Jerez for providing us with the HV-Inv software to model HVSRs using the diffuse wavefield theory. We thank Lukas
10 Preiswerk, Philippe Guéguen, and one anonymous reviewer for reviewing this manuscript.

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
