# Peer review of "Potentials and pitfalls of permafrost active layer monitoring using the HVSR method: A case study in Svalbard"

_Earth Surface Dynamics, 2018_

## Referee Comment (RC1) · L. Preiswerk (Referee) · 27 Jul 2018

**General comments**

Köhler and Weidle present results from two temporary seismic arrays installed in permafrost soil in Svalbard. They calculate horizontal-to-vertical spectral ratios (HVSR) for each station. At some stations, they observe a peak that glides to lower frequencies during the thawing period. They convincingly show, using simple numerical models, that this could correspond to a thawing layer with low shear wave velocity in the uppermost meters, addressing also potential issues due to the Nyquist frequency limit. They then give advice for future seismic projects in permafrost regions. Finally, they discuss whether an observed seismic tremor source could be used for HVSR studies.

This article is well-written and mostly clear. I do however have a number of specific comments for the text, for the organization of the manuscript, and especially for the figures. My main concern is the relation between HVSR from ambient noise and from the tremor source, which should be discussed in more detail to improve the coherency of the paper.

After minor revisions, I think that this paper would be an interesting contribution to the special issue in environmental seismology.

**Specific comments**

1. I am missing a paragraph with some explicit statements about how the HVSR of the ambient noise and of the tremor complement each other. Why would you need the tremor HVSR at all, what value does it add to noise HVSR? As is, section 6 is somewhat detached from the rest of the manuscript. By relating it better to the ambient noise HVSR, this part would be better integrated into the paper.
2. Organization: Sections 5, 6 and 7 would profit from a better structuring, i.e. clearly separating results from discussion, and giving recommendations only at the very end. Personally, I found the high number of enumerated lists and sublists confusing rather than helping. Please note that the order of these sections as stated at the end of section 1 is different to the order in the abstract and in the manuscript itself.
3. Figures 2, 3, A1, A2: I appreciate that you are showing HVSR from all stations. However, I did not get why the stations are ordered the way they are, and not e.g. ascending BRA1-8, and KBS1-4 (or similar). It took me some time to find the corresponding HVSR for stations mentioned in the text.
4. p.2 l.28 I suggest to better explain jargon (e.g. GSN, BH/HH channels, trigger mode) to the potential non-seismologists in the audience
5. p.4. l.1 does "North" refer to the spectrum or the time-domain record, i.e. do you average the raw data or the spectra?
6. p.4 l.4 Do you use Konno-Ohmachi smoothing? If so, please mention this and specify your smoothing constant. If not, please explain how you smooth your spectra.
   Konno, K., & Ohmachi, T. (1998). Ground-motion characteristics estimated from spectral ratio between horizontal and vertical components of microtremor. Bulletin of the Seismological Society of America, 88(1), 228–241.
7. p.7 l.4 You assume a 1-D subsurface, "inspired" by Haldorsen and Heim (1999). Could you please explain what each layer of your model corresponds to (e.g. regarding the units in Fig. 3+4 in Haldorsen and Heim (1999))? Why do you think that the 1-D assumption is justified given the clearly dipping layers?

8. p.8 Table 1: I think a figure would be much more helpful
9. p.8 l.5 Fäh et al. (2001) and Poggi et al. (2012) divide their spectra by a factor of sqrt(2) to compare the amplitudes. From Fig. 4 it looks like this would match quite well.
   Fäh, D., Kind, F., & Giardini, D. (2001). A theoretical investigation of average H/V ratios. Geophysical Journal International, 145(2), 535–549. https://doi.org/10.1046/j.0956-540x.2001.01406.x
   Poggi, V., Fäh, D., Burjánek, J., & Giardini, D. (2012). The use of Rayleigh-wave ellipticity for site-specific hazard assessment and microzonation: application to the city of Lucerne, Switzerland. Geophysical Journal International, 188(3), 1154–1172. https://doi.org/10.1111/j.1365-246X.2011.05305.x
10. p.8 l.15 wind: do you think that the wind directly affects the instruments, or do you think that the wind affects the ground which then is picked up by the geophones?
11. p.8 l.17 only wind noise or other noise as well?
12. p.9 l.8 This is the first time you mention tilt of the instruments. How did the instruments look like when they were dismantled, were they still leveled? I suggest to mention this in the Data section
13. p.9 l.9 Albaric et al. (in prep) does not appear in the references. Please elaborate or remove.
14. p.10 l.13 How deep is this concrete shelter, and how far away from the active layer?
15. p.11 l.13 amplitude spectrum: please rephrase by saying that you take the Fourier transform of these two time series (amplitude spectrum is technically correct, but a bit confusing in this context)
16. p.11 l.20 I am missing an actual physical mechanism of the tremor generation. Ocean waves have a lower frequency (in the microseism band) than the observed tremor. Why do you think that the cliff would vibrate at 4-5 Hz? What exactly would vibrate? What is the role of the cave, what would be this amplification (p.16 l.31)?
    p.11 l.23 "is a good explanation" I see that this phenomenon correlates with the tides, but in my opinion the source mechanism is not very clear, and should be discussed in more detail.
17. p.12 l.6 (and also in the summary p.14 l.16). Based on what test and significance criterion do you conclude that this is significant if it is within one standard deviation from the other?
18. p. 13 l.7 The horizontal and the vertical components are affected in the same way only if the source is a pure Rayleigh wave source.
19. p.15 l.15 What is the network code for KBS? Are there DOIs for the seismic datasets?
20. p.16 l.22 Only 31 tremors? In Fig. A3 I count at least 16 in one month.
21. p.17 l.23 I do not understand how the depth sensitivity plays a role, please elaborate.

**Figures:**

1. What is the source of the background image? Where is the borehole of Boike et al., 2018? There is a typo in Ny-Ålesund in panel b. Subtitle c is closer to panel b than panel c. It would be helpful to label the axes with North and East

2. In the top panel, it's almost impossible to distinguish red from dark red (same goes for Fig. 3 and those in the appendix)
3. see Fig. 2
4. This figure is quite busy. I suggest to make separate subplots with only the dashed lines, and subplots with only the solid lines and the same x-axis scale as d). Please also mention in the caption what the dashed lines show.
5. Why didn't you pick any peak frequencies in the end of July and in the beginning of August? I suggest to make more picks, and remove (or decrease the size of) the black dots from the figure, as you suggest that these are gliding peaks rather than discrete occurrences. Additionally, what line corresponds to which station? I recommend to plot the lines in different colors and make a legend.
7. In a), neither the "legend" nor the caption state whether dashed is summer or winter. In c), dark red and light red can hardly be distinguished. I am also missing a legend. As far as I understand, all RVSRs are from the tremor. If this is correct, please state so in the caption.

A3 It would be helpful to show the picks of your STA/LTA algorithm on this figure.

**Technical corrections**

p.1 l.4 (and several other places) thawn → thawed

p.2 l.15 it's → its (also p.10 l.16 and p.14 l.11)

p.2 l.20 anthropogentic → anthropogenic

p.5 l.23 extend → extent

p.6 l.5 theory,

p.7 l.2 S wave → S-wave

p.8 l.13 broadband

p.8 l.13 in and during beginning of the melt season → in the beginning and during the melt season?

p.9 l.13 kms$^{-1}$ → km s$^{-1}$

p.9 l.14 HVSR spectrum → HVSR curve?

p.9 l.23 relative → relatively

p.9 l.26 issues → issue

p.9 l.30 peaks/amplitude increase → peaks or amplitude increases

p.9 l.31 their → they

p.10 l.4 An → A

p.10 l.6 as well?

p.10 l.16 exist → exists

p.11 l.11 Weird wording and word order. Better to say e.g. like this: However, during neap tides and low wind speeds, some tidal maxima do not have a corresponding detection.

p.14 l.2+4 However used in two consecutive sentences

p.16 l.9 are → were

p.16 l.11 mis-detections → false positives

p.17 l.22 The lack of Rayleigh wave energy in this freq. band: which component are you talking about?

---

## Author Comment (AC1) · 8 Aug 2018

We very much appreciate the very detailed and thorough review provided by Lukas Preiswerk. We especially value the feedback concerning the structure of the paper. A response to all points raised by the reviewer can be found attached to this comment along with a revised, preliminary version of the manuscript highlighting all modifications.

Any further feedback is appreciated.

Please also note the supplement to this comment:

[Figure]

https://www.earth-surf-dynam-discuss.net/esurf-2018-52/esurf-2018-52-AC1-supplement.pdf

**Supplement:**

**Response to review of Lukas Preiswerk**

Andreas Köhler and Christian Weidle

August 6, 2018

We would like to thank Lukas Preiswerk for the very detailed and thorough review of our manuscript. We responded to all points raised by the reviewer below and attached a revised, preliminary version of the manuscript highlighting all modifications. We appreciate any further feedback.

**Specific comments**

1. I am missing a paragraph with some explicit statements about how the HVSR of the ambient noise and of the tremor complement each other. Why would you need the tremor HVSR at all, what value does it add to noise HVSR? As is, section 6 is somewhat detached from the rest of the manuscript. By relating it better to the ambient noise HVSR, this part would be better integrated into the paper.

We agree that the discussion of the results obtained from the tremor analysis can be better integrated into the manuscript. We therefore introduced a new discussion section. The old section 5, which focused only on ambient noise, has been renamed to "Discussion of the reliability of HVSRs for permafrost monitoring" and now includes the discussion of the tremor results. The part of former Section 6 describing the results of the tremor analysis has been moved before the new discussion section. We added a new paragraph in the discussion section about how tremor and ambient noise HVSRs complement each other in our case. KBS offers a much longer record of HVSR variability than our temporary network. However, KBS ambient noise cannot be used to directly resolve the H/V peak frequency caused by the active layer because of a too low sampling rate and lacking sensitivity inside the shelter to active layer changes. On the other hand, with the strong tremor signal at KBS, we are able to extract the Rayleigh wave ellipticities at low frequencies, which are presumably still affected by the very shallow structure. Therefore, we can resolve temporal variability over a time period of several years, which is most likely caused by the active layer.

2. Organization: Sections 5, 6 and 7 would profit from a better structuring, i.e. clearly separating results from discussion, and giving recommendations only at the very end. Personally, I found the high number of enumerated lists and sublists confusing rather than helping. Please note that the order of these sections as stated at the end of section 1 is different to the order in the abstract and in the manuscript itself.

We reorganized the order of sections (see also response above). Abstract and introduction have been updated accordingly. The numbered list in the discussion and the list of ambient noise HVSR observations have been removed and integrated into the text. However, we decided to keep the list of recommendation in the last section.

3. Figures 2, 3, A1, A2: I appreciate that you are showing HVSR from all stations. However, I did not get why the stations are ordered the way they are, and not e.g. ascending BRA1-8, and KBS1-4 (or similar). It took me some time to find the corresponding HVSR for stations mentioned in the text.

This was indeed a bit confusing. We reordered the stations in ascending order BRA1-8, and KBS1-4, from top to bottom in each figure. However, we choose to keep particular stations in the appendix since they do not add much information in the discussion, mainly because they do not exhibit a gliding frequency. Therefore, the stations do not appear in successive order in all four figures.

4. p.2 l.28 I suggest to better explain jargon (e.g. GSN, BH/HH channels, trigger mode) to the potential non-seismologists in the audience

The corresponding sentences have been rephrased.

5. p.4. l.1 does "North" refer to the spectrum or the time-domain record, i.e. do you average the raw data or the spectra?

The spectra of both horizontal components are averaged in the frequency domain. We rephrased to clarify.

6. p.4 l.4 Do you use Konno-Ohmachi smoothing? If so, please mention this and specify your smoothing constant. If not, please explain how you smooth your spectra. Konno, K., & Ohmachi, T. (1998). Ground-motion characteristics estimated from spectral ratio between horizontal and vertical components of microtremor. Bulletin of the Seismological Society of America, 88(1), 228-241.

A simple smoothing is done by convolving the spectrum with a boxcar function. Details such as smoothing length have been added in the text.

7. p.7 l.4 You assume a 1-D subsurface, "inspired" by Haldorsen and Heim (1999). Could you please explain what each layer of your model corresponds to (e.g. regarding the units in Fig. 3+4 in Haldorsen and Heim (1999))? Why do you think that the 1-D assumption is justified given the clearly dipping layers?8. p.8 Table 1: I think a figure would be much more helpful

We use a 1D structure inferred at the location of KBS from the geological crosssection provided by Haldorsen and Heim (1999). We obtained layer thickness from the figure and set seismic velocities typical for the rocks in the corresponding unit. We then modified the velocity structure iteratively to fit the observed and modeled tremor Rayleigh wave ellipticities. We agree that a 1D model might be too simple to explain the observations at tremor frequencies because of the presence of dipping layers. In fact, this might be another reason why we cannot exactly reproduce the measures tremor HVSRs, in addition to lacking knowledge about mode contribution. As described in the Appendix, we also attributed the discrepancy in tremor backazimuth and polarization to dipping layers. Modeling ellipticities using a 2D or 3D model is beyond the scope of the study, but could be a subject of future work. At higher frequencies (i.e., shorter wavelength and penetration depth), however, the sensitivity of HVSRs with respect to the area surrounding the measurement site is more confined. We therefore think that using a 1D model for modeling HVSRs with the diffuse wavefield theory is justified. We discuss those issues in the revised manuscript and added information about geological units in Table 1. We prefer presenting the velocity model in a Table because modeling is more reproducible by providing the exact values used in this study. Providing an additional figure would only add redundancy and unnecessarily increase the size of the manuscript in our opinion. However, if the reviewer thinks this to be absolutely necessary, we could provide another appendix figure.

9.  $p.8 \ l.5 \ F\ddot{a}h$  et al. (2001) and Poggi et al. (2012) divide their spectra by a factor of sqrt(2) to compare the amplitudes. From Fig. 4 it looks like this would match quite well.

According to Fäh et al. and Poggi et al., the normalization is needed when comparing HVSRs to Rayleigh wave ellipticities in case the quadratic mean of North and East is used to compute the horizontal component spectrum and an equal contribution from Rayleigh and Love waves is assumed. Therefore, this does not apply to the tremor since we compute radial to vertical spectral ratios and assuming pure Rayleigh waves on the radial component. In case of ambient noise HVSRs, we do not compare them with ellipticity, but with HVSRs modeled using the diffuse wavefield theory. However, we appreciate the comment since this made us reconsider how the mean horizontal spectrum is calculated in the diffuse wavefield code of Garcia-Jerez et al. (2016). In fact, the HVSR is defined as sqrt(2\*P1/P3) with P1 being the power on one horizontal component (e.g., East, and P1=P2) and P3 the power on the vertical component. However, we calculate HVSRs from our measurements using the geometric mean of Fourier amplitudes on the horizontal components: HVSR=sqrt(sqrt(P1)\*sqrt(P2)) / sqrt(P3). With the diffuse wavefield assumption, this is equivalent to HVSR = sqrt(P1/P3). Hence, the factor sqrt(2) arises here as well. We multiple our HVSRs with sqrt(2) in Fig 4c to make amplitude ratios comparable to the modeled ones.

10. p.8 l.15 wind: do you think that the wind directly affects the instruments, or do you think that the wind affects the ground which then is picked up by the geophones?

The shielding with gravel and rocks was supposed to reduce direct coupling with the wind. However, we cannot exclude that wind found its way through the rock pile and cause geophone vibration, especially after instruments lost good ground coupling. Therefore, it is probably a combination of both effects. Wind noise as such is of course not necessarily disturbing for HVSR measurement if seismic waves are excited at some distance to the receiver. However, in our case it seems that sources were close or at the installation site, so that the HV spectrum was affected and did not represent the site response. We rephrased the corresponding sentence.

11. p.8 l.17 only wind noise or other noise as well?

Noise in general. Sentence has been rephrased.

12. p.9 l.8 This is the first time you mention tilt of the instruments. How did the instruments look like when they were dismantled, were they still leveled?

**I suggest to mention this in the Data section**

Yes, some instruments were out of level and had to be re-leveled during maintenance. We added this information in the data section.

13. p.9 l.9 Albaric et al. (in prep) does not appear in the references. Please elaborate or remove.

This reference has been removed.

14. p.10 l.13 How deep is this concrete shelter, and how far away from the active layer?

In total, the shelter is about 2.5 to 3 m deep, hence not residing on the active layer, but surrounded by it. We added this information.

15. p.11 l.13 amplitude spectrum: please rephrase by saying that you take the Fourier transform of these two time series (amplitude spectrum is technically correct, but a bit confusing in this context)

We rephrased.

16.  $p.11\ l.20\ I$  am missing an actual physical mechanism of the tremor generation. Ocean waves have a lower frequency (in the microseism band) than the observed tremor. Why do you think that the cliff would vibrate at 4-5 Hz? What exactly would vibrate? What is the role of the cave, what would be this amplification ( $p.16\ l.31$ )?  $p.11\ l.23$  "is a good explanation" I see that this phenomenon correlates with the tides, but in my opinion the source mechanism is not very clear, and should be discussed in more detail.

While finding a quantitative physical model for the tremor source may be beyond the scope of this paper, we agree that the origin of the tremor can be explained and discussed in more detail. It is true that the source mechanism is not a direct coupling of water waves with the ground at longer periods (ocean microseism). We believe that slamming forces from breaking waves during cliff impact are a reasonable physical explanation. Similar phenomena have been observed and discussed in several studies. So-called high-frequency (HF) clifftop ground motion exhibit similar frequencies and temporal distribution (i.e. tidal modulation) as our observations. We added two sentences about this phenomenon and refer to a number of references for more details. We are not sure if and to what extent the cave plays a role, but since the tremor backazimuth points to the cave, there is probably a connection. The slamming forces of breaking ocean waves might be stronger in the cave because of the confined space.

17. p.12 l.6 (and also in the summary p.14 l.16). Based on what test and significance criterion do you conclude that this is significant if it is within one standard deviation from the other?

A standard Welch T test rejects the hypothesis of equal means at 99% confidence between 4 and 5.8 Hz. Information has been added.

18. p. 13 l.7 The horizontal and the vertical components are affected in the same way only if the source is a pure Rayleigh wave source.

The tremor generates Rayleigh as well as Love waves. However, we use the radial tremor component at KBS for the spectral ratio which exhibits a pure Rayleigh wave. Hence, the source magnitude does not affect the spectral ratios, and we should only observe the medium-dependent ellipticity. The sentence was rephrased.

19. p.15 l.15 What is the network code for KBS? Are there DOIs for the seismic datasets?

As stated in the data availability section, KBS data can be access through IRIS (DOI provided in reference list). Seismic data of the temporary network has not been made available yet at GFZ since the project has not finished yet. Data will eventually become available with a DOI such as for our previous project on Svalbard (see http://pmd.gfz-potsdam.de/gipp/showshort.php?id=escidoc:2850896). KBS network code has been added.

20. p.16 l.22 Only 31 tremors? In Fig. A3 I count at least 16 in one month.

Usually, more tremors are being detected during winter and autumn months. Fig. A3 shows a month of high activity. Between April and August (our deployment period in 2016), less tremors are observed which is typical for the spring and early summer season.

21. p.17 l.23 I do not understand how the depth sensitivity plays a role, please elaborate.

We removed this sentence.

Figures: 1. What is the source of the background image? Where is the borehole of Boike et al., 2018? Added in Fig.1 There is a typo in Ny-Ålesund in panel b. Subtitle c is closer to panel b than panel c. It would be helpful to label the axes with North and East

We added the source of satellite image in the caption, added the borehole position, and improved the figure formatting.

2. In the top panel, it's almost impossible to distinguish red from dark red (same goes for Fig. 3 and those in the appendix) 3. see Fig. 2

We changed soil temperature to a dashed line.

4. This figure is quite busy. I suggest to make separate subplots with only the dashed lines, and subplots with only the solid lines and the same x-axis scale as d). Please also mention in the caption what the dashed lines show.

We changed the x scale in (d) to 200 Hz. We would like to keep the scale for the modeled HVSRs to show theoretical peaks beyond the Nyquist frequency. We believe that dashed and solid lines should be plotted in the same subplot to better show the effect of the anti-aliasing filter and to avoid redundancy. However, we removed some curves for the sake of clarity: fewer models are plotted and HVSRs without contribution of Love waves are only shown in (c). We forgot to add dashed curve explanation in the caption.

5. Why didn't you pick any peak frequencies in the end of July and in the beginning of August? I suggest to make more picks, and remove (or decrease the size of) the black dots from the figure, as you suggest that these are gliding peaks rather than discrete occurrences. Additionally, what line corresponds to which station? I recommend to plot the lines in different colors and make a legend.

The figure has been modified. We picked more frequencies, removed the symbols and added gray scale for the stations.

7. In a), neither the "legend" nor the caption state whether dashed is summer or winter. In c), dark red and light red can hardly be distinguished. I am also missing a legend. As far as I understand, all RVSRs are from the tremor. If this is correct, please state so in the caption.

We modified the figure and added missing information.

A3 It would be helpful to show the picks of your STA/LTA algorithm on this figure.

To be honest, it would be quite complicated to add the detections in this plots since we used a special type of ObsPy plot and not a customized waveform plot that can be modified easily. Instead we would like to refer to Fig. 5a, where the STA/LTA picks are shown for a subset of this data section in the background of the time series of spectral amplitudes. We hope the reviewer finds this sufficient.

All technical corrections have been addressed.

[revised manuscript text omitted]

---

## Short Comment (SC1) · 15 Aug 2018

Comments on the paper entitled "Potentials and pitfalls of permafrost active layer monitoring using the HVSR method: A case study in Svalbard.", submitted for publication to Earth Surf. Dynam. Discuss. (#esurf-2018-52).

This manuscript is well-written, clear and it shows very convincing results on the correlation of the HVSR and external or internal factors (air temperature, wind, frozen soil layer etc...). The quality of the data, their processing and some relevant interpretation regarding the correlation (for example with ellipticity) provide evidences on the physical origins of HVSR variation.

I recommend this manuscript for publication. However, some critical lacks on the HVSR methods (a very classical methods used for very long time and benefiting from a huge experience and examples in scientific literature, references a little bit missing in this manuscript) and the irrelevant conclusions must be modified before publication.

1. On the HVSR - many recommendations related to the interpretation of the HVSR amplitude or the operative process for recording and processing HVSR, to the physical interpretation related to this method, in particular with ellipticity of Rayleigh waves, to the effect of the frozen uppermost layer on HVSR have been published for long time. I suggest the authors to browse these references and add them to their manuscript.

Ellipticity and HVSR - Lachet, C., & Bard, P. Y. (1994). Numerical and theoretical investigations on the possibilities and limitations of Nakamura's technique. *Journal of Physics of the Earth*, *42*(5), 377-397.
Method and processing - Chatelain, J. L., Guillier, B., Cara, F., Duval, A. M., Atakan, K., & Bard, P. Y. (2008). Evaluation of the influence of experimental conditions on H/V results from ambient noise recordings. *Bulletin of Earthquake Engineering*, *6*(1), 33-74.
Temperature and HVSR - Guéguen, P., Langlais, M., Garambois, S., Voisin, C., & Douste-Bacqué, I. (2017). How sensitive are site effects and building response to extreme cold temperature? The case of the Grenoble's (France) City Hall building. *Bulletin of earthquake engineering*, *15*(3), 889-906.
and more and more...

2. This lack reflects a lack of knowledge about this literature and may help authors to improve their manuscript. For example, the interpretation of the amplitude HVSR, the use of SPAC or FK methods for interpretation, the experimental condition recommendation, and other conclusion could be removed (very well known for very long time - see reference Chatelain et al. ) and it is not necessary to repeat them, just refer to already published papers.

I recommend the author to follow this recommendation before finalizing their manuscript.

Some additional remarks:
R1 As explain in these references, HVSR amplitude is poorly physically explained. In your manuscript, you focus more on the amplitude rather than on the value of the

frequency. Moreover, amplitude at high frequency is certainly controlled by local effects and for that reason, the amplitude may change quickly with local condition. What is happen at low frequency, i.e. between 1 and 10 Hz, that correspond approximately to the the uppermost layers of the soil? For using HVSR for environmental seismology, these frequency band must be considered in priority.

R2 Figure 7: do you think that the slight variation of ellipticity curves between winter and summer can be observed using HVSR? Please, provide uncertainties related to peaks.

R3. Figures 2 et 3 - Results are shown until months 8.5 and it is really a pity about the fact that we cannot see what's happen after. No more data?

R4: Model Tab. 1. How these parameters have been selected? Some other studies (in depth) about Vs and Vp in frozen regions have been published - Browse the very interesting journal on frozen region (Cox B, Wood C, Hazirbaba K (2012) Frozen and unfrozen shear wave velocity seismic site classification of Fairbanks, Alaska. J Cold Reg Eng 26(3):118–145. - Xu G, Yang ZJ, Dutta U, Tang L, Marx E (2011) Seasonally frozen soil effects on the seismic site response. J Cold Reg Eng 25(2):53–70.
and more and more...

---

## Author Comment (AC2) · 20 Aug 2018

We very much appreciate the comments given by Philippe Guéguen, in particular concerning the missing relevant literature. A response to all comments is provided attached to this comment along with a revised, preliminary version of the manuscript highlighting all modifications.

Please also note the supplement to this comment:
https://www.earth-surf-dynam-discuss.net/esurf-2018-52/esurf-2018-52-AC2-supplement.pdf

[Figure]

**ESurfD**

**Supplement:**

**Response to review of Philippe Guéguen**

Andreas Köhler and Christian Weidle

August 20, 2018

We would like to thank Philippe Guéguen for reviewing of our manuscript. We responded to all points raised by the reviewer below and attached a revised, preliminary version of the manuscript highlighting all modifications.

*1.On the HVSR - many recommendations related to the interpretation of the HVSR amplitude or the operative process for recording and processing HVSR, to the physical interpretation related to this method, in particular with ellipticity of Rayleigh waves, to the effect of the frozen uppermost layer on HVSR have been published for long time. I suggest the authors to browse these references and add them to their manuscript.*

*Ellipticity and HVSR - Lachet, C., & Bard, P. Y. (1994). Numerical and theoretical investigations on the possibilities and limitations of Nakamura's technique. Journal of Physics of the Earth,42(5), 377–397.*

*Method and processing - Chatelain, J. L., Guillier, B., Cara, F., Duval, A. M., Atakan, K., & Bard, P. Y. (2008). Evaluation of the influence of experimental conditions on H/V results from ambient noise recordings. Bulletin of Earthquake Engineering,6(1), 33-74.*

*Temperature and HVSR - Guéguen, P., Langlais, M., Garambois, S., Voisin, C., & Douste-Bacqué, I. (2017). How sensitive are site effects and building response to extreme cold temperature? The case of the Grenoble's (France) City Hall building. Bulletin of earthquake engineering, 15(3), 889-906.*

We absolutely agree that these references, especially the last one, are highly relevant for our study. Thanks for making us aware of these papers. We integrated the findings of these studies in the introduction, discussion and conclusion section.

*2. This lack reflects a lack of knowledge about this literature and may help authors to improve their manuscript. For example, the interpretation of the amplitude HVSR, the use of SPAC or FK methods for interpretation, the experimental condition recommendation, and other conclusion could be removed (very well known for very long time - see reference Chatelain et al. ) and it is not necessary to repeat them, just refer to already published papers.*

We agree that our list of recommendations did not clearly state which points have been addressed by previous studies. We now give, to our best ability, all the relevant references that concern observations and recommendations published previously. The results of Chatelain et al. are discussed in particular. However, we believe that our list of recommendations should also include issues raised by other studies, since this is a list concerning a particular application of HVSRs for active permafrost layer monitoring. We think that future experiments would benefit from such a compilation, even if parts are confirmations of previous findings. We re-phrased the conclusion to emphasize the new findings and known recommendation which we think are, however, of special importance in the context of permafrost monitoring.

*R1 As explain in these references, HVSR amplitude is poorly physically explained. In your manuscript, you focus more on the amplitude rather than on the value of the frequency. Moreover, amplitude at high frequency is certainly controlled by local effects and for that reason, the amplitude may change quickly with local condition. What is happen at low frequency, i.e. between 1 and 10 Hz, that correspond approximately to the the uppermost layers of the soil. For using HVSR for environmental seismology, these frequency band must be considered in priority.*

We do consider frequency peaks in case of the ambient noise HVSRs, i.e., the gliding peaks corresponding to the active layer. Amplitude features are discussed, however we agree, that they are hard to explain quantitatively and physically. We also agree (and described) that short-term variability depends on local site conditions. Hence, we do not draw conclusions from these properties concerning the sub-surface structure. Nevertheless, we do observe evidences for gliding peak frequencies for stations less affected by local conditions at frequencies higher than 10 Hz. We therefore believe that we provide enough evidence that environmental seismology can also utilize higher frequencies in the H/V spectrum.

In the second part of our paper, we present the results obtained for the tremor analysis. Here, we focus on the previously recommended frequency band between 1 and 10 Hz. Since we can extract the Rayleigh wave ellipticity from a dominant, directional source, we are confident that analyzing and interpreting changes in the HVSR (RVSR) amplitude is justified. We also confirm that these frequencies are still sensitive to the very shallow structure, although the resonance peak of the active layer is located at higher frequencies. We include the references suggested above to show that our findings are in line with previous studies.

*R2 Figure 7: do you think that the slight variation of ellipticity curves between winter and summer can be observed using HVSR? Please, provide uncertainties related to peaks.*

Using HVSRs computed from ambient seismic noise, we were not able to resolve the peak-trough structure and the seasonal variability that we found for the tremor ellipticity between 2 and 10 Hz. We believe that the sub-surface change in the active layer leads to an effect smaller than the uncertainties of the noise HVSRs in this frequency band. We are only able to resolve this feature with the dominant, repeating tremor signal.

The standard deviations of the tremor RVSRs are shown in Fig. 7 (now Fig 6). Since we do not pick peak frequencies in this case (only quantify amplitude deviations), no uncertainties are provided. However, maybe the comment refers to peak frequency uncertainties of the noise HVSRs / gliding peaks in Fig. 5 (now Fig. 7). We added the missing error bars for station BRA2.

*R3. Figures 2 et 3 - Results are shown until months 8.5 and it is really a pity about the fact that we cannot see what's happen after. No more data?*

We agree that a longer record would be desirable. However, for logistical reasons our field installations had to be recovered end of August / beginning of September in 2016. As also mentioned in the manuscript, the measurements were originally designed for a different purpose (glacier monitoring). We hope to repeat the experiment in future with a longer recording period.

*R4: Model Tab. 1. How these parameters have been selected? Some other studies (in depth) about Vs and Vp in frozen regions have been published - Browse the very interesting journal on frozen region (Cox B, Wood C, Hazirbaba K (2012) Frozen and unfrozen shear wave velocity seismic site classification of Fairbanks, Alaska. J Cold Reg Eng 26(3):118–145. - Xu G, Yang ZJ, Dutta U, Tang L, Marx E (2011) Seasonally frozen soil effects on the seismic site response. J Cold Reg Eng 25(2)*

We appreciate pointing us to these references. We selected model parameters for the deeper part from a local geological study. For Vs and Vp in frozen and unfrozen soil, we consulted the cited literature. We added the suggested references and updated velocity ranges in the text. Since we do not have direct body wave velocity measurements for the studied sites, we vary the seismic velocities in the active layer during modeling to cover a wide range of models.

[revised manuscript text omitted]

---

## Editor Comment (EC1) · F. Walter (Editor) · 18 Sep 2018

This Short Comment will now be treated as a Referee Comment, because the author had intended it as such.

---

## Referee Report (RR1)

```
from obspy import read, UTCDateTime
st = read("XX.079")
st += read("XX.080")

st.plot(type='dayplot',
        interval=24*60,
        color='k',
        tick_format='%Y-%m-%d',
        events=[{"time": UTCDateTime(2017,3,20,12)},
                {"time": UTCDateTime(2017,3,21,6)}])
```

---

## Referee Report (RR2)

**Review for "Potentials and pitfalls of permafrost active layer monitoring using the HVSR method: A case study in Svalbard" by Andreas Köhler and Christian Weidle**

The authors propose an interesting article about potentials and pitfalls of using the H/V method to monitor permafrost layers.
Obviously, the first pitfall is to use a correct definition of the H/V ratio. This is the main problem of the paper and it needs to be corrected before the scientific significance of the results can be assessed:
The authors use the geometric mean of the spectra of the northern and eastern component to calculate the H/V curves, which is not what is generally used for H/V calculations:

$$\frac{H}{V}(f) = \frac{\sqrt{|N(f)|^2 + |E(f)|^2}}{|Z(f)|}.$$

This formula has a physical meaning, i.e. the energies on both horizontal components are summed together and the resulting amplitude is taken for the H/V computation. In this way, waves from all azimuths are treated in the same way and a wave from any direction will always have the same H/V ratio.
I do not see a physical meaning in the formula used by the authors. For perfectly isotropic wave fields, both formulas might give the same result (with a sqrt(2) factor missing), but the authors especially state that the wave field is not isotropic here. Using the formula with the geometric mean, if the wave field was composed by a single Rayleigh wave, the resulting H/V curve would actually depend on the azimuth of the wave. Arriving from the direct north or east, one of the components would have a zero spectrum and therefore the so calculated H/V would be zero. Arriving from the northeast, both components would have the same amplitude and the H/V curve would be the correct one (except for a sqrt(2) factor).
For a real wave field, the case is more complex, and the spectra of the north and east components are sums of all wave contributions. Therefore, each wave will have a cos(alpha)*sin(alpha)*amplitude^2 contribution in the product N*E, plus the cross-terms between north and east components of the different waves. If the wave field is not isotropic, these cross-terms will in general not correct for the misestimation. Therefore, the basic definition for the H/V ratio used in this paper introduces a bias based on the main azimuths of the wave field. As the wave field in the area is not isotropic and changes with the seasons, this bias will result in a change of the H/V curve.
The real data H/V curves should be recalculated using the correct formula and the results should then be reinterpreted accordingly.

Apart from that, the study makes sense and is well described and performed. The identification of the tremor source is reasonable and sound. For the assessment of the variability of the Rayleigh wave ellipticity and the discussion section, the reinterpreted results might change the conclusions, so that it is difficult to comment on this issue at this time.
In any case, the study has or will have a high scientific value and be of great interest to the community.

Here are some other comments the authors might want to take into account:

Fig. 1   Maybe a photo of one or several of the stations would be helpful. The reader is curious to see how you install sensors in such an environment.

Fig. 2 and similar figures:     The plots are nice, but it is actually not that easy to see differences in the amplitudes in these plots. I would suggest adding the real H/V curves for comparison reasons.

5 HVSRs from a repeating seismic tremor:

I cannot follow the reasoning here. KBS sits at 2.5 m depth in a shelter. However, the wave field recorded by the station will still include the waves travelling through the frozen or not frozen layers. Waves with v=100 m/s at 20 Hz have a wavelength of 5 m, which are of course sensed by the station. Higher velocities result in larger wavelengths, so the station should be able to detect changes in the frozen layer in any case.

Appendix C and D are necessary for the understanding of the main text and are also cited so often in that text that they should be incorporated in the main document.

The verb "allow" usually requires a direct object (e.g.: "allows us to do …", "allows the determination of  …"; instead of "allows to do …").

---

## Author Response (AR2)

**Response to review of Lukas Preiswerk**

Andreas Köhler and Christian Weidle

November 26, 2018

We would like to thank Lukas Preiswerk for the additional comments. We responded to all points below and attached a revised version of the manuscript highlighting all modifications.

*- H/V and HVSR are used interchangeably, maybe it would be good to stick to HVSR to distinguish it from RVSR but avoiding confusion*

We replaced "H/V" with HVSR to be consistent throughout the entire manuscript.

*- p. 16, l.8 As expected, (comma missing)*

Corrected.

*- a few references do not have DOIs (Garcia-Jerez et al. (2016), both Haldorsen et al. papers, Jones et al. (2015))*

We added the missing DOIs.

*- Fig. A3: This looks to me like a dayplot from obspy. There, events can be easily marked by a star using a list of dictionaries containing the pick times. See attached example code and fig.*

Thank you very much for the advise. I should have studied the documentation of obspy more carefully.

**Response to reviewer 2**

Andreas Köhler and Christian Weidle

November 27, 2018

We would like to thank the reviewer for his helpful comments and suggestions. We responded to all points below and attached a revised version of the manuscript highlighting all modifications.

*The authors propose an interesting article about potentials and pitfalls of using the H/V method to monitor permafrost layers. Obviously, the first pitfall is to use a correct definition of the H/V ratio. This is the main problem of the paper and it needs to be corrected before the scientific significance of the results can be assessed: The authors use the geometric mean of the spectra of the northern and eastern component to calculate the H/V curves, which is not what is generally used for H/V calculations. ...*
*...The real data H/V curves should be recalculated using the correct formula and the results should then be reinterpreted accordingly.*

Thank you for addressing this issue in such detail. We absolutely agree with the reviewer that in case of an an-isotropic wavefield, it is physically correct to use the combined energy which is indeed the standard processing in H/V analysis. We re-calculated all H/V spectral ratios and updated all plots accordingly. Fortunately, none of our interpretations is affected and has to be changed. Using the correct equation simply leads to a change in the amplitude range (see new color scale ranges in updated figures), however the main features of the ambient noise HVSRs are consistent with the old results. The only significant change is visible in Fig. A3 for station BRA8. Using the old H/V definition resulted in unexplained low spectral ratios in June. This feature disappeared now. In Fig. 4 we removed the sqrt(2) factor and our H/V curves are now consistent with the forward modeling code.

*Apart from that, the study makes sense and is well described and performed. The identification of the tremor source is reasonable and sound. For the assessment of the variability of the Rayleigh wave ellipticity and the discussion section, the reinterpreted results might change the conclusions, so that it is difficult to comment on this issue at this time.*

In case of the tremor, we analyze only the radial-to-vertical spectral ratio. Hence, no averaging of the horizontal components is done, and all our interpretations and conclusions are valid.

*Fig. 1 Maybe a photo of one or several of the stations would be helpful. The reader is curious to see how you install sensors in such an environment.*

We added photos of the stations as an additional appendix figure.

*Fig. 2 and similar figures: The plots are nice, but it is actually not that easy to see differences in the amplitudes in these plots. I would suggest adding the real H/V curves for comparison reasons.*

We believe that using a color scale for the H/V amplitudes makes it easier to see and follow the change in peak frequency. Since we focus mainly on interpreting the peak frequencies instead of the amplitudes, we favor these plots. Examples of H/V curves are shown in Fig.4d for comparison. Plotting all H/V

curves (all days and all stations) in additional, would require introducing four new figures. We are unsure if adding a lot of redundant information would justify increasing the length of the paper. Nevertheless, we provide with this response figure versions that show H/V curves instead of a colored amplitude grid. We suggest to provide these figure only as supporting material, instead of including them in the paper or the appendix. We appreciate feedback if this is acceptable for the reviewer.

*5 HVSRs from a repeating seismic tremor: I cannot follow the reasoning here. KBS sits at 2.5 m depth in a shelter. However, the wave field recorded by the station will still include the waves travelling through the frozen or not frozen layers. Waves with v=100 m/s at 20 Hz have a wavelength of 5 m, which are of course sensed by the station. Higher velocities result in larger wavelengths, so the station should be able to detect changes in the frozen layer in any case.*

This was indeed a bit misleading. We agree that for example at 20 Hz the wave would still sense the material surrounding the shelter in summer and even more in winter, and we could therefore expect an effect on the measured H/V ratios. However, since the shelter area (sitting on frozen ground also in summer) is part of the medium the waves senses, the effect is probably lower than for a station placed directly on top of the soil. This could be the reason why we do not observed a clear seasonal effect at KBS close to the Nyquist frequency. We rephrased to clarify.

*Appendix C and D are necessary for the understanding of the main text and are also cited so often in that text that they should be incorporated in the main document.*

These parts were includeed in the main text in an earlier version. The reason why we decided to move these into the appendix, was to put more emphasis on the ambient noise HVSRs, and we were concerned that the detailed description of the tremor analysis in the main text would distract from the main focus of our paper, i.e., whether we can use HVSRs to monitor the active permafrost layer. However, we understand the concern of the reviewer and integrated Appendix C and D (and former Fig A6) back into the main text.

*The verb "allow" usually requires a direct object (e.g.: "allows us to do ...", "allows the determination of ..."; instead of "allows to do ...").*

Thank you for correcting this mistake. We changed the expressions in the manuscript accordingly.

[Figure]

Figure 1: Station BRA1,BRA2 and BRA4.

[Figure]

Figure 2: Station KBSA4,KBSA2 and BRA5.

[Figure]

Figure 3: Station BRA3,BRA6 and BRA7.

[Figure]

Figure 4: Station KBSA3,BRA8 and KBS.

[revised manuscript text omitted]